# Using Zebrafish to Screen Developmental Toxicity of Per- and Polyfluoroalkyl Substances (PFAS)

**DOI:** 10.3390/toxics12070501

**Published:** 2024-07-10

**Authors:** Katy N. Britton, Richard S. Judson, Bridgett N. Hill, Kimberly A. Jarema, Jeanene K. Olin, Bridget R. Knapp, Morgan Lowery, Madison Feshuk, Jason Brown, Stephanie Padilla

**Affiliations:** 1Oak Ridge Associated Universities Research Participation Program Hosted by EPA, Center for Computational Toxicology and Exposure, Biomolecular and Computational Toxicology Division, Rapid Assay Development Branch, U.S. Environmental Protection Agency, Research Triangle Park, NC 27711, USA; 2Center for Computational Toxicology and Exposure, Computational Toxicology and Bioinformatics Branch, Research Triangle Park, NC 27711, USA; rzjudson@gmail.com; 3Oak Ridge Institute for Science and Education Research Participation Program Hosted by EPA, Center for Computational Toxicology and Exposure, Biomolecular and Computational Toxicology Division, Rapid Assay Development Branch, U.S. Environmental Protection Agency, Research Triangle Park, NC 27711, USA; hill.bridgettn@gmail.com (B.N.H.); knapp.bridget@epa.gov (B.R.K.); 4Center for Public Health and Environmental Assessment, Immediate Office, U.S. Environmental Protection Agency, Research Triangle Park, NC 27711, USA; jarema.kimberly@epa.gov; 5Center for Computational Toxicology and Exposure, Biomolecular and Computational Toxicology Division, Rapid Assay Development Branch, U.S. Environmental Protection Agency, Research Triangle Park, NC 27711, USA; olin.jeanene@epa.gov (J.K.O.); mlowery.a@gmail.com (M.L.); 6Center for Computational Toxicology and Exposure, Scientific Computing and Data Curation Division, Data Extraction and Quality Evaluation Branch, U.S. Environmental Protection Agency, Research Triangle Park, NC 27711, USA; feshuk.madison@epa.gov; 7Center for Computational Toxicology and Exposure, Scientific Computing and Data Curation Division, Application Development Branch, U.S. Environmental Protection Agency, Research Triangle Park, NC 27711, USA; brown.jason@epa.gov

**Keywords:** PFAS, perfluorooctanesulfonamide, larval zebrafish, in vivo, gross morphology, benchmark concentration (BMC), potency, high-throughput screening

## Abstract

Per- and polyfluoroalkyl substances (PFAS) are found in many consumer and industrial products. While some PFAS, notably perfluorooctanoic acid (PFOA) and perfluorooctanesulfonic acid (PFOS), are developmentally toxic in mammals, the vast majority of PFAS have not been evaluated for developmental toxicity potential. A concentration–response study of 182 unique PFAS chemicals using the zebrafish medium-throughput, developmental vertebrate toxicity assay was conducted to investigate chemical structural identifiers for toxicity. Embryos were exposed to each PFAS compound (≤100 μM) beginning on the day of fertilization. At 6 days post-fertilization (dpf), two independent observers graded developmental landmarks for each larva (e.g., mortality, hatching, swim bladder inflation, edema, abnormal spine/tail, or craniofacial structure). Thirty percent of the PFAS were developmentally toxic, but there was no enrichment of any OECD structural category. PFOS was developmentally toxic (benchmark concentration [BMC] = 7.48 μM); however, other chemicals were more potent: perfluorooctanesulfonamide (PFOSA), N-methylperfluorooctane sulfonamide (N-MeFOSA), ((perfluorooctyl)ethyl)phosphonic acid, perfluoro-3,6,9-trioxatridecanoic acid, and perfluorohexane sulfonamide. The developmental toxicity profile for these more potent PFAS is largely unexplored in mammals and other species. Based on these zebrafish developmental toxicity results, additional screening may be warranted to understand the toxicity profile of these chemicals in other species.

## 1. Introduction

Per- and polyfluoroalkyl substances (PFAS) are a class of synthetic compounds used in everyday items, including cookware, textiles, food packaging, and electronics [1,2,3]. In the last decade, attention has been paid to PFAS due to their ubiquitous presence in environmental matrices, such as soil, water (ground and surface), as well as human blood (maternal and fetal cord), contributing to a better understanding and delineation of PFAS-related toxicity to human health and the environment [1,4,5,6,7]. The Organisation for Economic Co-operation and Development (OECD) has recently characterized PFAS as having at least one fully fluorinated methyl or methylene carbon without any H/Cl/Br/I atoms attached to it [8]. The fluorinated methyl or methylene bond imparts properties that render PFAS as generally possessing high stability, low reactivity, and varying levels of bioactivity [2]. Estimates of the number of PFAS in the environment vary depending on the PFAS definition applied. As per the Toxic Substances Control Act (TSCA) 8(a)(7) rule (EPA, 2024), there are approximately 13,000 PFAS (which can be represented by a discrete chemical structure). It is estimated that ~650 are included as part of the non-confidential TSCA inventory and are still actively being produced and used in commercial products, with an unknown number of degradation products and manufacturing byproducts [9].

While much is still unknown about the adverse effects of PFAS, a subset of legacy compounds, namely perfluorooctanesulfonic acid (PFOS), perfluorooctanoic acid (PFOA), and related perfluoroalkyl acids (PFAAs) [10,11], have been well-studied, leading to the publication of lifetime drinking water health advisories for selected PFAS [12]. Many of the health effects linked to PFAS exposure were also associated with some legacy compounds, including changes in immune and thyroid function, reproductive challenges, liver disease, and cancer [10]. This extensive range of effects could be due to PFAS exposures varying by structure, route, or duration. In addition, the bioaccumulation potential of these compounds could lead to differences in internal doses for both human and environmentally relevant species [7,10,13,14]. A recent systematic review of the PFAS literature focusing on mammalian toxicological and epidemiological studies revealed that one of the primary targets of PFAS toxicity was developmental processes, and that there is an overall poor understanding of developmental toxicity potential for the majority of PFAS included in that review [15]. Although these reviews and mammalian studies have advanced the understanding of PFAS-related human adverse effects, only a small proportion of PFAS have any in vivo toxicity data. Thus, there is a need for rapid, high-throughput, screening tools to prioritize this diverse set of chemicals, especially for developmental toxicity.

Zebrafish (*Danio rerio*), small freshwater fish native to Southeast Asia, have been utilized in many toxicological studies as a model for developmental toxicity assessments (e.g., [16,17,18,19]). Zebrafish share many developmental signaling pathways, organ systems, metabolism, and brain structure/functions with mammals [20,21,22,23,24], leading to the relatively high concordance of developmental toxicity outcomes with other vertebrates [16,19,25,26]. There has been an increase in zebrafish PFAS investigations for developmental toxicity, but most of the research has only focused on a few PFAS (selected examples, [27,28,29,30,31]). Recently, however, larger PFAS chemical screens have utilized early life-stage zebrafish. One study, using dechorionated embryos exposed for five days to 139 PFAS, showed developmental and neurodevelopmental toxicity linked to chemical volatility and structural features, suggesting that grouping these chemicals may aid in identifying toxicity [32]. Similarly, it has been reported [33] that, by varying exposure windows and duration (4–72 h), a selection of PFAS chemicals (*n* = 38) induced developmental toxicity, in particular hepatotoxicity, with lipid transport potentially playing a role in these observations. Another screening effort testing 74 PFAS in developing zebrafish [34] associated chemical structural features to bioconcentration factors and metabolic pathways in larvae exposed to either 0.5 or 5.0 μM of each PFAS. Despite using different experimental designs and endpoints, perfluorooctanesulfonamide (PFOSA) was among the most potent developmentally toxic PFAS across these studies [32,33,34].

In 2019, the U.S. Environmental Protection Agency (EPA) began a coordinated research effort to screen a large PFAS library using an array of different in vitro and in vivo high- and medium-throughput toxicity assays to inform chemical category and read-across approaches [35]. As part of that screening effort, an in vivo, developing zebrafish model was employed to assess the developmental toxicity of 182 unique PFAS from the EPA PFAS chemical library. Various landmarks of development in zebrafish larvae were assessed following PFAS developmental exposure, and the results were combined with previously published information on the chemical purity assessments of the stocks used to treat the zebrafish [36]. This type of quality control (QC) check at this stage of experimentation has not been included in all the PFAS screens mentioned above and, given that a significant number (55/182: 30%) of the chemicals failed this basic stock quality assessment, analytical chemical QC could be a significant confounder of data interpretation. The present results were also compared to previously published data on the developmental effects of PFAS in zebrafish embryos/larvae to gain a better understanding of the manner in which these chemicals affect zebrafish in various developmental assays.

## 2. Materials and Methods

### 2.1. Experimental Animals

All research and breeding procedures in this study were reviewed and approved by the Office of Research and Development’s Health Institutional Animal Care and Use Committee (IACUC) at the U.S. EPA in Research Triangle Park, NC (Protocol #21-08-003; approved 8 August 2018, and Protocol #24-09-002; approved 2 September 2021). The animal facility is an internationally accredited Association for Assessment and Accreditation of Laboratory Animal Care (AAALAC) facility (Unit# 000509). The parental fish were wild-type adult zebrafish (*Danio rerio*) descended from an undefined outbred stock originally supplied by both Aquatic Research Organisms, Hampton, NH, and EkkWill Waterlife Resources, Ruskin, FL, USA. The adult zebrafish were maintained at a density of 7 fish/L in 3.5 L tanks and housed in recirculating zebrafish housing racks (Tecniplast USA, West Chester, PA, USA) with reverse osmosis-purified tap water (Durham, NC, USA), which was buffered with Instant Ocean Sea Salt (Spectrum Brands, Blacksburg, VA, USA) and sodium bicarbonate (Church & Dwight, Co., Ewing, NJ, USA). The water was maintained at 28 °C, pH 7.4, and conductivity (1000 μS/cm), with ammonia and nitrite (maintained at 0 ppm) and nitrate (allowed in insignificant amounts). The fish were fed twice a day with decapsulated artemia (E-Z Egg; Brine Shrimp Direct, Ogden, UT, USA) and Gemma Micro 300 formulated diet (Skretting, Westbrook, ME, USA). The housing rooms were illuminated according to a 14:10 h light:dark cycle (lights on at 07:00 h). For embryo production, groups of approximately 150 same-age mixed sex zebrafish (ages ranging from 3 to 15 months old) were moved into 16 L on rack recirculating spawning tanks (Z-Park tanks, Tecniplast USA, West Chester, PA, USA) about one week before embryos were needed. Then, on the afternoon before embryos were needed, mesh spawning platform inserts were added. Embryos were collected the following morning approximately 45 min after the room lights came on (07:45 h) and were maintained at 28 °C for 1 to 2 h until washing. A diagram of the experimental methods is included in Figure 1.

### 2.2. Embryo Rearing

The embryos were placed in 600 mL beakers and kept at 28 °C until washing [37] two times with 0.06% bleach (*v*/*v*) in 10% Hanks’ Balanced Salt Solution (13.7 mM NaCl, 0.54 mM KCl, 25 μM Na_2_HPO_4_, 44 μM KH_2_PO_4_, 130 μM CaCl_2_, 100 μM MgSO_4_, and 420 μM NaHCO_3_ [hereinafter referred to as Hanks’]) for 5 min each wash, then rinsed with Hanks’ alone after each bleach wash. Immediately following washing, the embryos were examined, and healthy embryos were separated from dead or unfertilized eggs and moved into fresh Hanks’.

### 2.3. Chemical Exposure

All PFAS were received from Evotec Inc. (Branford, CT, USA) at concentrations ranging from 5 to 30 mM, solubilized in dimethyl sulfoxide (DMSO). All researchers were blinded to the chemical identity until after all data were collected and analyzed. Stock plates were first prepared with the highest concentration of each chemical, dependent on their level of initial solubilization provided by Evotec. The final concentration of the DMSO vehicle in every well was 0.4% (*v*/*v*). A single-concentration screening approach was implemented whereby the highest concentration of each chemical was tested to determine which PFAS were likely to be positive for developmental toxicity. A positive chemical was liberally defined, selecting those where 2/6 (33.3%) or more of the larvae per condition appeared to be affected either by death and/or malformations. An additional multiple-concentration screening of the positive chemicals (*n* = 87) was then conducted along with ≥15% (*n* = 13; PFAS selected by a random number generator) of the chemicals that were negative in the single-concentration screen.

The multiple-concentration PFAS screening plates were made up with the highest concentration of each positive chemical as well as the chosen negatives and then were serially diluted with DMSO to produce an 8-point concentration–response curve (half-log dilution interval). Each experimental plate for both the single high-concentration test (described above) and the multiple concentration–response assessment contained ≥22 vehicle control wells (DMSO; >99.9% purity; Sigma-Aldrich (St. Louis, MO, USA); Chemical Abstract Services [CAS] number 67-68-5; 0.4% *v*/*v* final concentration) and 2 positive control wells (both containing chlorpyrifos [97% purity; Sigma-Aldrich]); CAS number 2921-88-2; 30 μM final concentration; extensive historical data show that this chlorpyrifos concentration will cause either death or severe malformations in zebrafish embryos by 6 dpf. A typical stock plate held nine PFAS as part of the 8-point concentration response curve and was used to dose six identical plates for each group of nine chemicals. Since the plates contained an arrangement with a single well for each concentration of each of the nine chemicals, six identical plates dosed from that same stock plate that produced a total *n* = 6 wells per chemical per dose. All final exposure concentrations for each chemical are listed in Appendix A, Col F of the second sheet.

Between 6 and 8 h post-fertilization (hpf), healthy embryos (with chorions) were transferred, one embryo per well, into 96-well (0.5 mL) microtiter plates (Cell Culture-Treated, Flat-Bottom Microplate 96 well [Corning™ Costar™, Kennebunk, ME, USA; Cat # 09-761-145]) filled with 200 μL of Hanks’ solution. A random number generator was used to assign the order in which the rows of embryos were plated in the 96-well plates each week. After plating, the embryos were dosed with 0.8 μL of the chemical dosing solution, then immediately sealed with AlumaSeal II™ (Excel Scientific Inc., Victorville, CA, USA) to prevent the volatilization of the chemicals, and then placed in a leakproof secondary container. These containers were placed in an incubator maintained on a 14:10 h light:dark cycle at 26 °C for rearing. On 5 dpf, live larvae were gently moved out of the plate with test chemicals and into a new 96-well mesh plate (Millipore Corp., Bedford, MA, USA), just with the Hanks’ solution. Once within the mesh well plate, the animals were rinsed with 400 μL of fresh Hanks’ three times, and then the plates were covered with a non-adhesive material (Microseal^®^ A, BioRad, Hercules, CA, USA), the plate lid was added, and then the plate sides were wrapped in Parafilm™ (PM992, Bermis Company, Neenah, WI, USA) and returned to the incubator. This rinse on 5 dpf was conducted to lessen the possible exposure of the human assessors to the chemicals during the detailed examination of each larva on 6 dpf.

### 2.4. Larval Assessments

On 6 dpf between 8 and 10 AM, two individual assessors, blinded to the chemical treatments, independently examined each larva for mortality, hatching status, and malformations using an Olympus SZH10 stereo microscope. Mortality was defined as a lack of heartbeat or presence of coagulation. Malformations were defined as uninflated swim bladder, craniofacial defects, edema, spinal defects, decreased pigmentation, abnormal position in water column, tail defects, or blood pooling (some examples can be viewed in Figure 1; data are in Appendix A). If more than 15% of the negative control larvae were abnormal (i.e., dead, not hatched, or malformed) or if the positive control larvae were not at least 50% abnormal, that plate was not used for any analysis. Over the course of the experiment, there were no plates that needed to be removed from analysis based on the criteria above. Additionally, the overall rate of normal animals in the controls for the entire study was 97% (Appendix A). After all plates were assessed, the larvae were anesthetized using cold shock, and then euthanized with a cold 20% bleach solution.

### 2.5. Concentration–Response Modeling

The raw data from the larval assessments consisted of counts of larvae observed at each of the twelve endpoints: living (dead or alive), hatching (hatched or unhatched), swim bladder (non-inflation), craniofacial defects (dysmorphology of the head or eyes), edema, spinal defects (curved spine), pigmentation, position (in water column, either persistent lying on one side or upside down), tail defects (e.g., kinks), or blood pooling (Appendix A). Each assessor also assigned every larva a general ranking for the overall condition: normal (no defects present), abnormal (defects present), or severely abnormal (life-threatening defects present). Observations were recorded for each chemical and concentration. If any of these defects were present in a larva, the additional endpoint “any” was set to 1.

Concentration–response data were processed using the ToxCast Data Analysis Pipeline (tcpl) R-package (tcpl v.3.2.0; https://cran.r-project.org/web/packages/tcpl/index.html; accessed on 4 March 2024 [38,39]). These endpoints are anticipated to be released in Fall 2024 with ToxCast’s InvitroDB v4.2 at https://doi.org/10.23645/epacomptox.6062623 (accessed on 8 April 2024). For the endpoint-chemical per concentration index, counts were aggregated to a percentage, called endpoint scores, with dead larvae excluded. For example, if there were 5 (out of 6) living larvae and 2 had edema at a tested concentration, the edema score would equal 2/5 × 100 = 40%. No additional normalization was performed, and outliers were not excluded. For each endpoint-chemical pair, the concentration–response series was fit to 5 bounded models (constant, hill, gain–loss, exponential 4–5), with the winning model selected by the lowest Akaike Information Criteria (AIC) score, a statistical calculation that compares model quality. Unbounded models available in tcpl were excluded given the dichotomous nature of observations. To estimate activity, a cutoff threshold was set at 16% and a continuous hit call (hitc) value was determined as the product of the following components: (1) at least one median response greater than the assay cutoff threshold, (2) the maximal efficacy in the fitted response is larger than the assay cutoff, and (3) the AIC score of the winning model is less than the constant model [40]. Classification criteria for continuous hit calls were set in line with other in vitro screening efforts as: hitc = 0 as negative, 0 < hitc < 0.9 as equivocal, and hitc ≥ 0.9 as positive [41]. In addition to the estimated activity concentrations inducing a specified level of responses (e.g., 10%, 50%, etc.), a benchmark concentration (BMC) was also derived in tcpl using a specified benchmark response level (BMR) of 1.349 times the standard deviation of the baseline (10%) [42]. Given the lack of baseline variability in the dichotomous observations, the baseline median absolute deviation was set as 5%.

## 3. Results

The raw data collected for each larva at 6 days post-fertilization (dpf) from each assessor are shown in Appendix A. These data then underwent concentration–response modeling (described in detail in Section 2) for the twelve developmental endpoints, and benchmark concentrations (BMCs) were calculated for each positive endpoint/chemical. All the BMC graphs for the positive endpoints for each chemical are presented in Appendix A. The calculations and results for the BMC determinations are presented in Appendix A. Figure 2 shows an example of the best fit BMC dose–response modeling results with a description of the information provided on each graph. In the example, the chemical is PFOSA, and the endpoint is swim bladder non-inflation (“score.swim_bladder”). The black circles represent the fraction of larvae that were positive for that morphological endpoint at the given concentration in micromolar (noted on the *x*-axis; “1e+00” ≡ 1 × 10^0^ ≡ 1 μM). The black Xs are the fraction of animals that were alive and morphologically normal at that concentration; in this example, 6/6 animals at the four highest concentrations (3, 10, 30, and 100 μM) were dead at 6 dpf. The black curve is the best fit model to the concentration–response data, and the “mthd” listed at the top is the name of best fit model function (all possible functions are shown in Figure 2 of Ref. [38]). The gray band surrounding the *x*-axis indicates the background noise cutoff level.

A total of 185 PFAS chemicals (182 unique) were assessed for developmental toxicity in zebrafish embryos/larvae, which included measurements of malformation or mortality. Of those 185 chemicals, 56 (30%) caused developmental toxicity. Three PFAS were tested twice and were highly reproducible: 1,6-diiodoperfluorohexane tested positive both times, with potencies within an order of magnitude; and pentafluoropropanoic anhydride and 1H,1H-perfluorooctylamine, both of which tested negative both times. These duplicate chemicals are highlighted in yellow in Appendix A. Therefore, ignoring the duplicates, 55/182 (30%) PFAS caused developmental toxicity.

One strength of the present study is that the same stock chemicals tested in the zebrafish were also assessed for chemical presence and purity [36]. Chemicals that passed quality control (QC) were the ones where the compound was present by molecular match; fragmentation patterns putatively confirmed structure; peak area percentage was >85% of the total response; and/or instrument attenuation was less than the threshold based on signal-to-noise ratio or general peak height. A chemical passed the quality control check if it was detected with analytical instrumentation according to the above criteria and failed if either it was not detected or if degradation had occurred. Note that concentration was not determined for most of the chemical stocks, primarily because standards were not available. In the few situations where a standard was available and the concentration was estimated (Table 5 of Ref. [36]), many of the chemicals were only 60% to 90% of their target concentration. 

If the quality control data for the chemical stocks were combined with the zebrafish developmental data (Table 1 or Figure 3), 127 (70%) of the 182 unique chemicals passed the quality control. Considering only the 55 chemicals that affected zebrafish development, 49 of those chemicals passed the quality control. Interestingly, six of the chemicals that affected zebrafish development failed the quality control. This suggests that the toxicity may have been caused by an unknown substance or a mixture of the parent PFAS and its degradation products. Table 1 indicates the chemicals that failed the quality control labeled with an “F” and the ones that passed labeled with a “P”. In Figure 3, the chemicals that failed the chemical quality control are designated by one of three symbols, “⊗” or “x” or “Δ”; “⊗” shows the chemicals that failed the QC and did not result in developmental toxicity. “x” represents the chemicals that failed the quality control and were also likely to be volatile (vapor pressure exceeded 100 Tor [100 mm Hg]). The high volatility indicates that the chemical was more likely to be lost during the solubilization and/or testing process. The six chemicals that failed the quality control but were positive for eliciting developmental toxicity are designated with an open triangle (Δ) in Figure 3. These six chemicals were removed from further consideration in subsequent analyses. The chemicals that did not cause developmental toxicity are noted in Table 1 or Appendix A as having a BMC of “1000”; chemicals positive for developmental toxicity have a numerical BMC listed with values equal to or less than 100 μM, the highest concentration tested.

Figure 3 visualizes the toxicity of the 182 unique chemicals across OECD structural categories [43] to assess whether there are PFAS categories that were more or less likely to be developmentally toxic. The lowest BMC was used in this figure for each chemical; the specific endpoint that was used is noted in Appendix A (Column K). The large colored squares with the cartoon of a fish inside represent the PFAS that produced developmental toxicity; the color relates to the potency (i.e., BMC; yellow is the least potent and red is the most potent). Almost any class with three or more chemicals that passed the chemical quality control contained at least one positive chemical; so, it is difficult to declare whether a particular chemical class is more or less likely to contain developmentally toxic chemicals. The three most potent chemicals were PFOSA (BMC = 0.26 μM), N-MeFOSA (BMC = 0.44 μM), and ((Perfluorooctyl)ethyl)phosphonic acid (BMC = 0.58 μM), all of which had BMCs of less than 1 μM and represent three different structural classifications: PFAA precursors, other aliphatics, and FASA-based PFAA precursors. There were 18 PFAS with BMCs less than 10 μM representing nine different classes, and 28 PFAS with a BMC between 10 and 100 μM representing seven different classes. The highest chemical concentration tested was 100 μM; therefore, all BMC results for positive chemicals fall below that threshold (Table 1 and Figure 3). Interestingly, the sulfonamide structure seemed especially toxic to the developing zebrafish as five out of the six sulfonamide containing PFAS chemicals that also passed the QC produced developmental toxicity (Table 1). All five of these sulfonamides in the zebrafish were markedly toxic having a BMC below 20 μM, whereas only one did not and was considered a negative. That negative sulfonamide (CAS number 1691-99-2; N-ethyl-N-(2-hydroxylethyl) perflurorooctanesulfonamide) has the same basic structure as the most potent sulfonamide (PFOSA; CAS number 754-91-6), except with more extensive functional groups attached to the sulfonamide moiety. Perhaps these extra ethyl and hydroxylethyl functional groups block the toxic action of the sulfonamide group in some manner or impair its uptake into the larva. The apparent toxicity of the sulfonamide structure was reinforced by a ToxPrint chemotype enrichment analysis [35,44,45], showing that three structures were significantly associated with developmental toxicity in the zebrafish embryos/larvae: sulfonamide, sulfonyl, or an 8-carbon chain length (the results are in column L of Appendix A). The sulfonamide structure is a subset of the sulfonyl structure, i.e., a chemical cannot be classified as a sulfonamide without also being classified as a sulfonyl, but the opposite is not necessarily true. The sulfonamide association does not account for all the sulfonyl-associated toxicity as there were sulfonyl containing PFAS that were not sulfonamides but were also active.

A commonly used general method to assess the influence of structure on the toxicity of PFAS chemicals is to compare the chain length of the PFAS versus its likelihood to cause toxicity. For the PFAS that passed the QC, a longer chain length was associated with a higher likelihood of developmental toxicity: if the chain length of the toxic PFAS is compared to the chain length of the non-toxic PFAS, the average chain length of the toxic PFAS was significantly longer than the chain length of the non-toxic PFAS (*p* = 0.035) (Figure 4A). There was, however, no correlation between the chain length and the increased potency of the chemical (r^2^ = 0.24; a *p*-value for slope = 0.28; *n* = 49) (Figure 4B). When the degree of potency (BMC) is plotted against the chain length for each chemical that produced developmental toxicity, no relationship was found (Figure 4B); therefore, the potency of the chemical does not tend to change as the chain length increases.

The evidence of the lack of a significant relationship between OECD structural groups and the toxicity profile is also supported by the heat maps depicted in Figure 5. In this case, the potency of each positive endpoint for each chemical that passed the QC is indicated with the color representing the degree of potency. This was performed to discern any patterns of specific endpoints that might be associated with the different PFAS structural classifications. The left panel (A) shows the independent hierarchical clustering of the chemicals by the pattern of endpoint affected to evaluate if any of the chemical categories (listed by color in the lower right legend) are enriched in certain endpoints or endpoint patterns. As can be seen by the extremely varied representation of the PFAS category colors in the far-left column, there does not appear to be any pattern of effects related to a particular PFAS structural category. This is further explained by the results shown in the righthand panel (Figure 5B) showing the pattern of effect by PFAS category, again visualizing that any one category (B, left column) does not present with a signature profile of endpoints.

## 4. Discussion

Although there did not appear to be any OECD structural groups that were more or less likely to contain chemicals positive for developmental toxicity, another type of analysis revealed that the sulfonamide containing PFAS were especially likely to be developmentally toxic to zebrafish and possibly other vertebrates. Like other reports [46,47,48], there was a general tendency for the longer chain compounds to be toxic, but that did not extend to a direct relationship between potency and chain length.

The QC testing of the chemical stocks was a definite advantage in this study as substances could volatilize during chemical handling, such as those with higher vapor pressure, which could be interpreted as false negatives for developmental toxicity. Given that a large percentage of the PFAS chemicals failed the QC, it seems irresponsible for future PFAS screens to proceed without analytical QC. Despite this library being selected for its structural diversity [35], in order to identify chemical grouping(s) with a higher rate of positives, there is still additional testing of the PFAS chemicals that will be needed to inform health and environmental hazard and enable accurate read-across comparisons for this class of compounds.

This study evaluated a large group of PFAS to determine developmental toxicity during early-life exposure. As part of the analysis, the results were compared to those from other PFAS library screening studies (Table 2). Notably, three screens using a zebrafish developmental model with similar assessment methods were reviewed [32,33,34], with results compared in Table 2. Differences in experimental design choices among laboratories, including rearing temperature, chorion status, dosing range, windows of exposure, and statistical methods for potency determination, made direct comparisons difficult among the studies. Even with those caveats, some common trends can be discerned in the results. There were fifteen chemicals that were tested in all four laboratories. Of those fifteen, fourteen passed the QC based on the testing associated with the present study. Of all chemicals, PFOSA was the most potent in every laboratory. PFOS (CAS number 754-91-6) was also tested in all four laboratories, but only tested positive for developmental toxicity in two out of the four laboratories, though this comparison is confounded by dosing range choices in each laboratory. In all four laboratories, PFOS was less potent than PFOSA. Although PFOSA is metabolized to PFOS in zebrafish [34] and other species [49], PFOSA has a markedly higher bioaccumulation factor than PFOS [34], which could increase the time × tissue concentration of PFOSA and may help to explain why the parent chemical is more toxic than the metabolite. Even though PFOSA was so potent across multiple laboratory settings in larval zebrafish, no PFOSA developmental toxicity studies in mammals were identified. Using the Expanded PFAS Evidence Map Dashboard [50], only two studies of PFOSA in mammals were identified: both involved acute dosing and resulted in evidence of systemic effects in rats, but neither contained any assessment of developmental toxicity. Therefore, it may be important going forward to focus on more developmental toxicity testing among chemicals such as PFOSA. It should be noted that, in the present study as well as in the other studies listed in Table 2, the endpoints employed could best be classified as gross morphological changes. Normal-appearing animals do not necessarily signal the lack of developmental toxicity for the test chemical. Internal anatomy, physiology, and function may have been affected by the PFAS exposure without affecting gross morphology.

It has been widely reported that PFAS exposure causes changes in the thyroid axis in adults and developing animals. In a recently published meta-analysis of 13 human studies, a correlation between PFAS exposure and changes in maternal or newborn thyroid function was noted [51], while in an analysis of National Health and Nutrition Examination Survey (NHANES) data, an association between PFAS exposure and thyroid disorders was also revealed [52]. In a separate study of a smaller population of mothers and newborns, it was noted that the newer “replacement” PFAS disrupt newborn thyroid status as much as or more than the legacy PFAS, like PFOS [53]. Zebrafish possess a thyroid axis very similar to the mammalian thyroid axis [23,54,55,56,57,58] and are sensitive to the developmental effects of many confirmed thyroid-disrupting chemicals (reviewed in [24]). Likewise, as a major regulatory organ, the disruption of thyroid axis development in zebrafish may cause malformations or alter physiology and behavior during development. Mirroring results in mammals, zebrafish exposed to PFAS show disrupted thyroid development and function: changes in thyroid hormone levels and/or of thyroid-related genes have been noted in zebrafish after exposure to PFAS [59,60,61,62,63]. Given this previous literature, the present study’s morphological analysis concentrated on the known effects of thyroid hormone disruption, like decreased pigmentation and disrupted swim bladder inflation (reviewed in [64]). The results do not reveal any significant changes in pigmentation (Appendix A), but many of the chemicals (31 out of the 49; 63%) that were positive for developmental toxicity resulted in an uninflated swim bladder (Figure 5), with that effect being one of the most sensitive endpoints. The effects of PFAS chemicals on zebrafish swim bladder non-inflation have been reported by others [28,65]. As noted above, this effect of PFAS on swim bladder inflation could be due to an indirect effect through the perturbation of the thyroid axis, or it could possibly be due to the direct action of the PFAS on the surfactant lipid profile of the swim bladder, which is necessary for normal structure and function [66]. Failure to inflate could be related to problems with surfactant lipid function as it has been noted in a study of the effects of PFAS in human bronchial epithelial cells [67] or general membrane disruption [68]. While the present study did not assess the mechanistic underpinnings of the swim bladder non-inflation or possible thyroid disrupting effects, this could be a promising area for future work. To that end, the comparison of the present results on zebrafish development with a recently published in vitro thyroid screen [69] using many of the same chemicals tested in this paper revealed some possible insights into mechanism. One hundred percent of the PFAS that affected human or Xenopus iodotyrosine deiodinase (hIYD or xIYD) also affected development in the zebrafish embryos. Although zebrafish do possess IYD [70], the authors [69] of the in vitro screening study felt that the IYD enzymes were not a “sensitive” target of the tested PFAS. Perhaps zebrafish deiodinases are more sensitive to PFAS inhibition than the human or Xenopus counterparts, or the consequences of enzyme inhibition during development are more significant than anticipated.

In general, longer chain PFAS are more lipophilic, less water soluble, and tend to have longer half-lives, which is thought to contribute to their increased toxicity [46,47,48]. It was observed that whether a PFAS was developmentally toxic or not was related to chain length, but these data were quite variable. For the 49 chemicals assessed as positive for developmental toxicity, there was no direct relationship between potency and chain length, i.e., that the increase in chain length was not associated with an increase in toxicity. This lack of a clear trend between chain length and potency mirrors results from other PFAS zebrafish developmental toxicity studies [28] as well as other developmental in vitro studies [44]. It, however, contrasts with results of clinical chemistry endpoints in rodent subchronic oral toxicity studies using PFAS of various chain lengths [71], where the chain length was shown to be associated with the health outcomes for PFAS. At least in developing vertebrates, there may be more structural variables besides chain length contributing to the toxicity profile. We also found a lack of association between OECD structural categories and developmental toxicity potential, which was also observed in a study of developmental neurotoxicity in vitro assays [44]. The absence of structure-based bioactivity relationships may be explained by limitations in biologically informative groupings of PFAS structures and ongoing work towards reproducible PFAS structure-based chemical categories, which may further inform toxicity [35]. An alternate chemotyping analysis [35,44,45] showed that three substructures were associated with zebrafish developmental toxicity: sulfonyl, sulfonamide, and an 8-carbon chain length; the sulfonamide structure has also been highlighted by others exploring PFAS developmental toxicity [29]. In addition, there was a relationship between developmental toxicity in zebrafish and certain PFAS as has been noted previously in other zebrafish in vivo [32,33,34] and in vitro [44,72] PFAS screening studies.

This study of developmental toxicity in larval zebrafish using a diverse structural library of PFAS has illustrated the usefulness of this type of medium-throughput screening for potency and possible effects related to chemical structure. Despite no clear pattern regarding structural features related to toxic or non-toxic PFAS using the OECD structural groupings of the 182 chemicals, other structural identifiers were noted. Finally, leveraging the analytical QC results for the PFAS chemicals proved useful and should be encouraged in future studies, not only to eliminate the possibility for false negatives or positives, but also to promote the standardization of methodologies across laboratories.

## Figures and Tables

**Figure 1 toxics-12-00501-f001:**
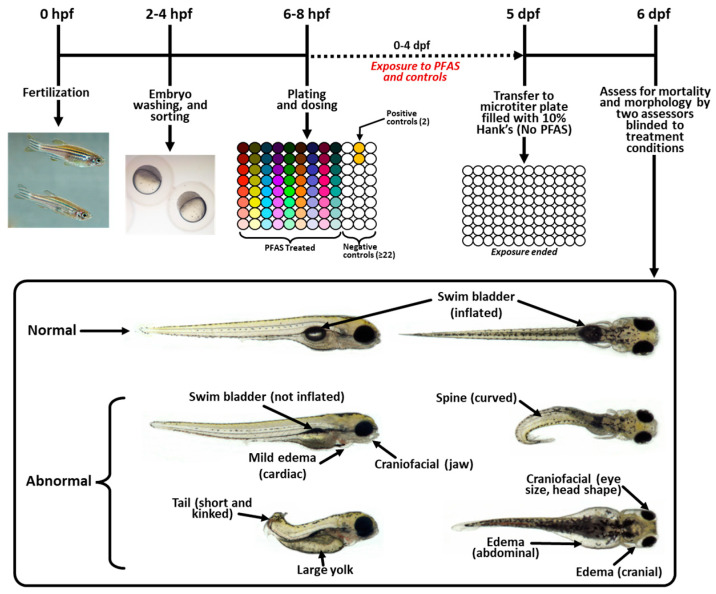
**Experimental design:** Experimental events from 0 h post-fertilization (hpf) through 6 days post-fertilization (dpf). Embryo washing, plating, and chemical exposure occurred on 0 dpf. Each column of colored circles on the 6–8 hpf plate represents a different chemical. Morphological assessments were conducted on 6 dpf by two experimenters blinded to treatment group information. Images show a normal embryo as well as several of the common developmental malformations identified in this analysis.

**Figure 2 toxics-12-00501-f002:**
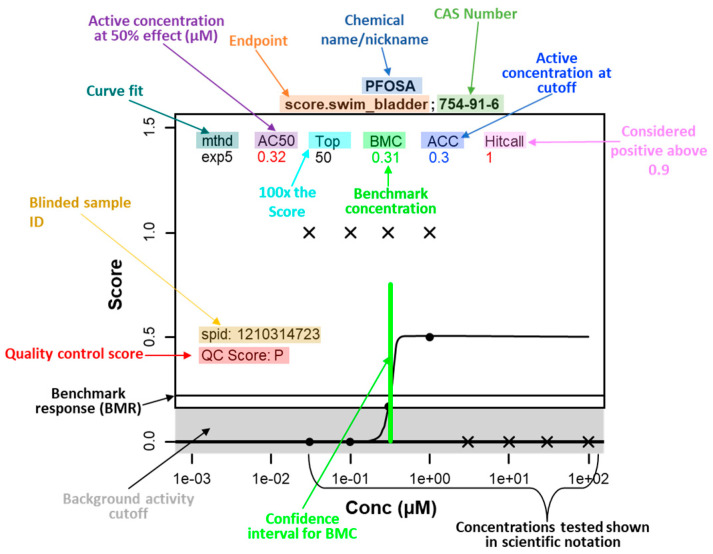
**BMC determination example:** Concentration–response data for a given chemical and endpoint are presented in this example; the chemical name is noted in the first line of the title, and the endpoint and CAS numbers are noted in the second line of the title. In the example, the chemical is PFOSA, and the endpoint is swim bladder non-inflation (“score.swim_bladder”). Score refers to percentage of animals affected divided by 100. The black circles represent the fraction of larvae that were positive for that morphological endpoint at the given concentration in micromolar (noted on the *x*-axis; “1e+00” ≡ 1 × 10^0^ ≡ 1 μM). The black Xs are the fraction of larvae that were alive and morphologically normal at that concentration; in this example, all 6/6 animals at the four highest concentrations (3, 10, 30, and 100 μM) were dead at 6 dpf. The black curve is the best fit model to the concentration–response data, and the “mthd” listed at the top is the name of best fit model function (all possible functions are shown in Figure 2 of Ref. [38]). The gray band near the bottom of the graph (0–0.16) indicates the background noise cutoff level. The horizontal line above the cutoff is the benchmark response (BMR) and is equal to 1.349 * cutoff of 0.16 [42]. For an active response (i.e., hitcall), the point at which the BMR intersects the model curve is the benchmark concentration (BMC, μM), indicated by the vertical green line, with the width representing the 95% confidence interval. In the lower left corner of each graph, the sample ID (spid) and the analytical quality control (QC) score for that sample (P = pass or F = fail: [36]) are shown. For more information on curve fitting visit https://clowder.edap-cluster.com/files/659c5239e4b063812d5d00cc?dataset=64b81ac2e4b08a6b5a3c2528&space=647f710ee4b08a6b394e426b (accessed on 6 May 2024).

**Figure 3 toxics-12-00501-f003:**
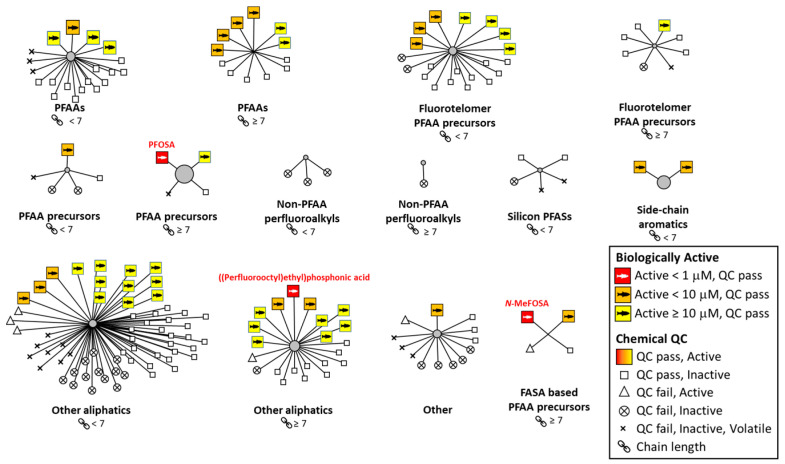
**Visualization of toxicity, QC verdict, and structural class:** This diagram summarizes and compares the biological activity and chemical quality control (QC) status as a function of OECD PFAS structural category [43]. Each group is one chemical category separated by chain length groups of those less than 7 (<7) or those greater than or equal to 7 (≥7). The lengths of the lines from the center of each radial plot were arbitrarily selected for visual spacing purposes and confer no additional information. Each point is one chemical representing the 182 unique ones tested. A large square is assigned to a chemical that was active in at least one endpoint, and its color indicates the potency of the most active endpoint. Each of the three most potent chemicals is indicated by a red chemical name above its respective red square. Small squares indicate an inactive chemical that passed the QC. Chemicals that failed the QC are broken down into three groups based on volatility and activity. PFAA = perfluoroalkyl acid, PFAS = perfluoroalkyl substance, FASA = fluoroalkyl sulfonamide.

**Figure 4 toxics-12-00501-f004:**
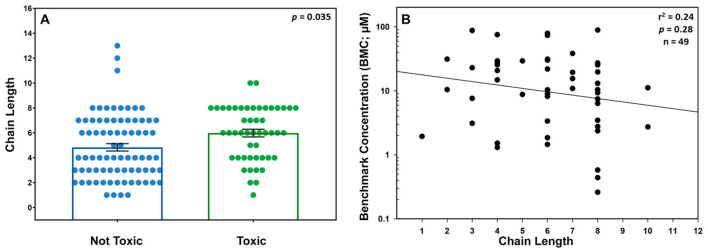
**Comparison of toxicity and chain length:** (**A**) Bar graph showing the relationship between the chain length and developmental toxicity of the chemical. Each dot represents one unique chemical. Only chemicals that passed the QC are included (*n* = 49 green dots for toxic [has a BMC] and *n* = 78 blue dots for non-toxic [no concentration relation or BMC] chemicals). A Mann–Whitney U non-parametric test showed that there was a significant difference (*p* = 0.0035) in average chain length between the two groups. (**B**) Linear regression showing that an increase in chain length for the positive chemicals was not associated with increased toxicity. r^2^ = 0.24; a *p*-value for slope = 0.28; *n* = 49.

**Figure 5 toxics-12-00501-f005:**
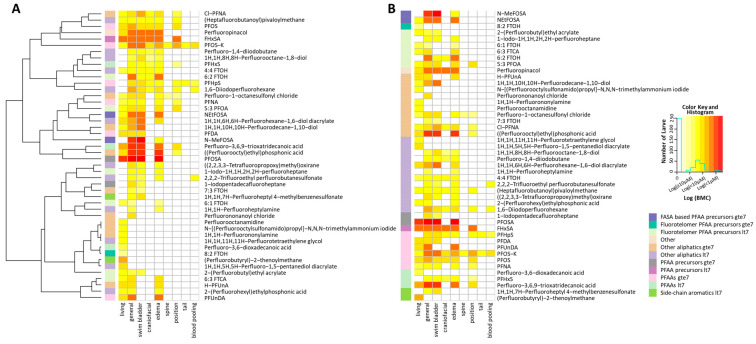
**Heat maps showing the relationships among specific endpoints and structural class:** Colors (upper right legend) indicate (BMC; μM), with deeper red colors indicating increased potency. Only chemicals that passed the analytical quality control (QC) and were active in at least one endpoint are shown. Left panel (**A**): Chemicals are sorted by unsupervised hierarchical clustering to discern if any of the chemical categories (listed by color in the lower right legend) are enriched in certain endpoints. Right panel (**B**): Endpoint enrichment is also explored with data sorted by chemical category.

**Table 1 toxics-12-00501-t001:** **Chemical list and properties:** Each of the unique 182 chemicals tested are listed one per row and include (from left to right) chemical name; an abbreviation or synonym, if appropriate; structural category; CAS number; DTXSID (which links directly to the CompTox dashboard; https://comptox.epa.gov/dashboard/; accessed on 8 February 2024); molecular weight; formula; chain length; quality control data (QC) verdict [36]; vapor pressure (mm Hg) OPERA model values from https://comptox.epa.gov/dashboard/ (accessed on 16 March 2024); and benchmark concentration (BMC) in micromolar (μM). The table is sorted by BMC from the lowest value to the highest.

Chemical Name	Abbreviation/Synonym	Structural Category	CAS Number	DTXSID	Molecular Weight	Formula	Chain Length	QC	Vapor Pressure (mm Hg)	BMC (μM)
Perfluorooctanesulfonamide	PFOSA	PFAA precursors gte7	754-91-6	DTXSID3038939	499.14	C8H2F17NO2S	8	P	2.45 × 10^−1^	0.26
*N-Methylperfluorooctanesulfonamide*	N-MeFOSA	FASA based PFAA precursors gte7	31506-32-8	DTXSID1067629	513.17	C9H4F17NO2S	8	P	1.20 × 10^−4^	0.44
((Perfluorooctyl)ethyl)phosphonic acid		Other aliphatics gte7	80220-63-9	DTXSID30627108	528.10	C10H6F17O3P	8	P	1.27 × 10^−4^	0.58
Perfluoro-3,6,9-trioxatridecanoic acid		PFAAs lt7	330562-41-9	DTXSID50375114	562.08	C10HF19O5	4	P	6.61 × 10^−5^	1.30
Perfluorohexanesulfonamide	FHxSA	PFAA precursors lt7	41997-13-1	DTXSID50469320	399.13	C6H2F13NO2S	6	P	2.49 × 10^−4^	1.45
1H,1H,6H,6H-Perfluorohexane-1,6-diol diacrylate		Other aliphatics lt7	2264-01-9	DTXSID80379721	370.20	C12H10F8O4	4	P	5.06 × 10^−3^	1.52
1,6-Diiodoperfluorohexane		Other aliphatics lt7	375-80-4	DTXSID90190949	553.86	C6F12I2	6	P	2.91 × 10^−1^	1.85
Perfluoropinacol		other	918-21-8	DTXSID60238701	334.06	C6H2F12O2	1	P	1.41 × 10^0^	1.94
*N-Ethylperfluorooctane* sulfonamide	NEtFOSA	FASA based PFAA precursors gte7	4151-50-2	DTXSID1032646	527.20	C10H6F17NO2S	8	P	5.02 × 10^−6^	2.37
Perfluoroundecanoic acid	PFUnDA	PFAAs gte7	2058-94-8	DTXSID8047553	564.09	C11HF21O2	10	P	6.48 × 10^−4^	2.74
Potassium perfluorooctanesulfonate	PFOS-K	PFAAs gte7	2795-39-3	DTXSID8037706	538.22	C8F17KO3S	8	P	2.48 × 10^−6^	2.77
(Perfluorobutyryl)-2-thenoylmethane		Side-chain aromatics lt7	559-94-4	DTXSID7060332	322.20	C10H5F7O2S	3	P	4.23 × 10^−2^	3.11
2-(Perfluorohexyl)ethanol	6:2 FTOH	Fluorotelomer PFAA precursors lt7	647-42-7	DTXSID5044572	364.11	C8H5F13O	6	P	6.01 × 10^−1^	3.37
1H,1H,10H,10H-Perfluorodecane-1,10-diol		Other aliphatics gte7	754-96-1	DTXSID50369896	462.13	C10H6F16O2	8	P	3.14 × 10^−2^	3.47
1-(Perfluorofluorooctyl)propane-2,3-diol		Other aliphatics gte7	94159-84-9	DTXSID80881157	494.15	C11H7F17O2	8	F	1.27 × 10^−2^	4.42
9-Chloro-perfluorononanoic acid	Cl-PFNA	Other aliphatics gte7	865-79-2	DTXSID30382104	480.53	C9HClF16O2	8	P	1.17 × 10^−2^	6.38
*N-Methyl-N-*(2-hydroxyethyl)perfluorooctanesulfonamide	N-MeFOSE	FASA based PFAA precursors gte7	24448-09-7	DTXSID7027831	557.22	C11H8F17NO3S	8	F	1.01 × 10^−5^	7.06
Perfluorooctanesulfonic acid	PFOS	PFAAs gte7	1763-23-1	DTXSID3031864	500.13	C8HF17O3S	8	P	2.48 × 10^−6^	7.48
1H,1H,5H,5H-Perfluoro-1,5-pentanediol diacrylate		Other aliphatics lt7	678-95-5	DTXSID5060986	320.19	C11H10F6O4	3	P	1.67 × 10^−2^	7.62
2,2,3,3-Tetrafluoropropyl acrylate		Other aliphatics lt7	7383-71-3	DTXSID10224331	186.11	C6H6F4O2	2	F	3.51 × 10^0^	7.81
3-(Perfluorohexyl)propanoic acid	6:3 FTCA	Fluorotelomer PFAA precursors lt7	27854-30-4	DTXSID70379917	392.12	C9H5F13O2	6	P	1.06 × 10^−1^	8.20
2H,2H,3H,3H-Perfluorooctanoic acid	5:3 PFOA	Fluorotelomer PFAA precursors lt7	914637-49-3	DTXSID20874028	342.11	C8H5F11O2	5	P	4.39 × 10^−1^	8.77
1H,1H,7H-Perfluoroheptyl 4-methylbenzenesulfonate		Side-chain aromatics lt7	424-16-8	DTXSID30340244	486.27	C14H10F12O3S	6	P	1.95 × 10^−4^	8.97
3H-Perfluoro-2,2,4,4-tetrahydroxypentane		other	77953-71-0	DTXSID70379295	262.08	C5H5F7O4	1	F	3.89 × 10^−8^	9.29
Perfluorodecanoic acid	PFDA	PFAAs gte7	335-76-2	DTXSID3031860	514.09	C10HF19O2	8	P	1.46 × 10^−3^	9.34
1H,1H,8H,8H-Perfluorooctane-1,8-diol		Other aliphatics lt7	90177-96-1	DTXSID30396867	362.12	C8H6F12O2	6	P	1.18 × 10^−1^	10.37
Perfluoro-1-octanesulfonyl chloride		Other aliphatics gte7	423-60-9	DTXSID90315130	518.57	C8ClF17O2S	8	P	1.75 × 10^1^	10.42
((2,2,3,3-Tetrafluoropropoxy)methyl)oxirane		Other aliphatics lt7	19932-26-4	DTXSID70880230	188.12	C6H8F4O2	2	P	3.13 × 10^0^	10.44
1-Iodopentadecafluoroheptane		PFAA precursors gte7	335-58-0	DTXSID5059828	495.96	C7F15I	7	P	1.04 × 10^2^	10.89
11-H-Perfluoroundecanoic acid	H-PFUnA	Other aliphatics gte7	1765-48-6	DTXSID5061954	546.10	C11H2F20O2	10	P	1.37 × 10^−3^	11.17
Perfluorononanoyl chloride		Other aliphatics gte7	52447-23-1	DTXSID00379925	482.52	C9ClF17O	8	P	2.33 × 10^2^	13.10
Perfluoro-1,4-diiodobutane		Other aliphatics lt7	375-50-8	DTXSID30190948	453.84	C4F8I2	4	P	3.32 × 10^1^	14.80
Perfluoroheptanesulfonic acid	PFHpS	PFAAs gte7	375-92-8	DTXSID8059920	450.12	C7HF15O3S	7	P	3.33 × 10^−7^	15.55
Perfluorooctanamidine		Other aliphatics gte7	307-31-3	DTXSID70381151	412.10	C8H3F15N2	7	P	3.12 × 10^−1^	19.45
*N-*[(Perfluorooctylsulfonamido)propyl]-*N*,*N*,*N*-trimethylammonium iodide		Other aliphatics gte7	1652-63-7	DTXSID8051419	726.23	C14H16F17IN2O2S	8	P	1.17 × 10^−4^	19.75
2-(Perfluorobutyl)ethyl acrylate		Fluorotelomer PFAA precursors lt7	52591-27-2	DTXSID1068772	318.14	C9H7F9O2	4	P	9.39 × 10^−1^	20.74
1H,1H-Perfluoro-3,6,9-trioxadecan-1-ol		Other aliphatics lt7	147492-57-7	DTXSID40380797	398.08	C7H3F13O4	2	F	1.64 × 10^−2^	21.25
2-(Perfluorohexyl)ethylphosphonic acid		Other aliphatics lt7	252237-40-4	DTXSID20179883	428.09	C8H6F13O3P	6	P	3.40 × 10^−6^	21.90
(Heptafluorobutanoyl)pivaloylmethane		Other aliphatics lt7	17587-22-3	DTXSID3066215	296.19	C10H11F7O2	3	P	1.01 × 10^−1^	22.97
Perfluorononanoic acid	PFNA	PFAAs gte7	375-95-1	DTXSID8031863	464.08	C9HF17O2	8	P	8.44 × 10^−3^	25.45
4:4 Fluorotelomer alcohol	4:4 FTOH	Other aliphatics lt7	3792-02-7	DTXSID60377821	292.15	C8H9F9O	4	P	6.28 × 10^0^	25.53
1H,1H-Perfluorononylamine		Other aliphatics gte7	355-47-5	DTXSID50379930	449.11	C9H4F17N	8	P	9.67 × 10^−2^	27.23
Perfluoro(2-(2-propoxypropoxy)-1H,1H-propan-1-ol)		Other aliphatics lt7	14548-74-4	DTXSID80371164	482.09	C9H3F17O3	3	F	1.43 × 10^−3^	27.37
1H,1H,5H-Perfluoropentyl methacrylate		Other aliphatics lt7	355-93-1	DTXSID90880131	300.15	C9H8F8O2	4	P	4.39 × 10^−1^	27.89
1-Iodo-1H,1H,2H,2H-perfluoroheptane		Fluorotelomer PFAA precursors lt7	1682-31-1	DTXSID9061881	424.00	C7H4F11I	5	P	8.54 × 10^−1^	29.34
2,2,2-Trifluoroethyl perfluorobutanesulfonate		Other aliphatics lt7	79963-95-4	DTXSID60380390	382.12	C6H2F12O3S	4	P	3.01 × 10^1^	29.36
Perfluorohexanesulfonic acid	PFHxS	PFAAs lt7	355-46-4	DTXSID7040150	400.11	C6HF13O3S	6	P	8.19 × 10^−9^	30.15
1H,1H,11H,11H-Perfluorotetraethylene glycol		Other aliphatics lt7	330562-44-2	DTXSID00380798	410.11	C8H6F12O5	2	P	7.85 × 10^−6^	31.27
Potassium perfluorohexanesulfonate	PFHS-K	PFAAs lt7	3871-99-6	DTXSID3037709	438.20	C6F13KO3S	6	P	8.19 × 10^−9^	31.30
7:3 Fluorotelomer alcohol	7:3 FTOH	Other aliphatics gte7	25600-66-2	DTXSID50382621	428.14	C10H7F15O	7	P	3.00 × 10^−1^	38.32
6:1 Fluorotelomer alcohol	6:1 FTOH	Fluorotelomer PFAA precursors lt7	375-82-6	DTXSID00190950	350.08	C7H3F13O	6	P	5.05 × 10^−1^	73.35
Perfluoro-3,6-dioxadecanoic acid		PFAAs lt7	137780-69-9	DTXSID50381073	446.07	C8HF15O4	4	P	3.42 × 10^−3^	75.28
1H,1H-Perfluoroheptylamine		Other aliphatics lt7	423-49-4	DTXSID10379835	349.10	C7H4F13N	6	P	5.01 × 10^−1^	79.17
3:3 Fluorotelomer carboxylic acid	3:3 FTCA	Fluorotelomer PFAA precursors lt7	356-02-5	DTXSID00379268	242.09	C6H5F7O2	3	P	5.19 × 10^−1^	87.79
2-(Perfluorooctyl)ethanol	8:2 FTOH	Fluorotelomer PFAA precursors gte7	678-39-7	DTXSID7029904	464.12	C10H5F17O	8	P	2.07 × 10^−1^	89.03
(Heptafluoropropyl)trimethylsilane		Silicon PFASs lt7	3834-42-2	DTXSID70400078	242.21	C6H9F7Si	3	F	1.28 × 10^2^	1000
(Perfluoro-5-methylhexyl)ethyl 2-methylprop-2-enoate		Fluorotelomer PFAA precursors lt7	50836-66-3	DTXSID60379901	482.19	C13H9F15O2	4	P	1.13 × 10^−2^	1000
(Perfluorobutyl)ethene		Fluorotelomer PFAA precursors lt7	19430-93-4	DTXSID6047575	246.08	C6H3F9	4	F	2.14 × 10^3^	1000
(Perfluoroheptyl)methyl methacrylate		Other aliphatics gte7	3934-23-4	DTXSID5063235	468.16	C12H7F15O2	7	P	4.03 × 10^0^	1000
(Perfluoropropyl)methyl methacrylate		Other aliphatics lt7	13695-31-3	DTXSID3065586	268.13	C8H7F7O2	3	P	1.19 × 10^0^	1000
1-(Perfluorohexyl)octane		Fluorotelomer PFAA precursors lt7	133331-77-8	DTXSID20440585	432.27	C14H17F13	6	P	2.67 × 10^−3^	1000
1,1,1,3,3-Pentafluorobutane		Other	406-58-6	DTXSID5073901	148.08	C4H5F5	1	F	2.81 × 10^3^	1000
1,1,1,5,5,5-Hexafluoro-2,4-pentanedione		Other	1522-22-1	DTXSID4061753	208.06	C5H2F6O2	-	-	2.81 × 10^2^	1000
1,6-Dibromododecafluorohexane		Other aliphatics lt7	918-22-9	DTXSID20335129	459.86	C6Br2F12	6	P	8.38 × 10^1^	1000
11:1 Fluorotelomer alcohol	11:1 FTOH	Fluorotelomer PFAA precursors gte7	423-65-4	DTXSID80375107	600.12	C12H3F23O	11	P	4.16 × 10^−3^	1000
1-Bromopentadecafluoroheptane		Other aliphatics gte7	375-88-2	DTXSID9059919	448.96	C7BrF15	7	F	2.93 × 10^2^	1000
1H,1H,2H,2H-Perfluorohexyl iodide		Fluorotelomer PFAA precursors lt7	2043-55-2	DTXSID1047578	373.99	C6H4F9I	4	P	5.38 × 10^0^	1000
1H,1H,2H-Perfluoro-1-decene		Fluorotelomer PFAA precursors gte7	21652-58-4	DTXSID7074616	446.11	C10H3F17	8	F	4.07 × 10^2^	1000
1H,1H,5H-Perfluoropentanol		Other aliphatics lt7	355-80-6	DTXSID0059879	232.07	C5H4F8O	4	P	2.85 × 10^1^	1000
1H,1H,7H-Dodecafluoro-1-heptanol		Other aliphatics lt7	335-99-9	DTXSID9059832	332.09	C7H4F12O	6	P	6.18 × 10^−1^	1000
1H,1H,8H,8H-Perfluoro-3,6-dioxaoctane-1,8-diol		Other aliphatics lt7	129301-42-4	DTXSID70381090	294.10	C6H6F8O4	2	P	1.72 × 10^−4^	1000
1H,1H,9H-Perfluorononyl acrylate		Other aliphatics gte7	4180-26-1	DTXSID00194615	486.15	C12H6F16O2	8	P	7.06 × 10^−1^	1000
1H,1H-Heptafluorobutyl epoxide		Other aliphatics lt7	1765-92-0	DTXSID10379254	226.09	C6H5F7O	3	P	1.58 × 10^2^	1000
1H,1H-Perfluorooctyl acrylate		Other aliphatics gte7	307-98-2	DTXSID5059799	454.14	C11H5F15O2	7	P	6.31 × 10^0^	1000
1H,1H-Perfluorooctylamine		Other aliphatics gte7	307-29-9	DTXSID50184723	399.10	C8H4F15N	7	P	1.06 × 10^−1^	1000
1H,1H-Perfluoropentylamine		Other aliphatics lt7	355-27-1	DTXSID60377826	249.08	C5H4F9N	4	F	4.22 × 10^1^	1000
1H,2H-Hexafluorocyclopentene		Other aliphatics lt7	1005-73-8	DTXSID10461880	176.06	C5H2F6	3	F	5.15 × 10^2^	1000
1H-Perfluoro-1,1-propanediol		Other aliphatics lt7	422-63-9	DTXSID9059969	166.05	C3H3F5O2	2	F	1.41 × 10^−1^	1000
1-Iodo-1H,1H,2H,2H-perfluorononane		Fluorotelomer PFAA precursors gte7	2043-52-9	DTXSID90880156	524.01	C9H4F15I	7	P	1.14 × 10^−1^	1000
1-Pentafluoroethylethanol		Other aliphatics lt7	374-40-3	DTXSID70880134	164.08	C4H5F5O	2	F	3.95 × 10^1^	1000
1-Propenylperfluoropropane		Fluorotelomer PFAA precursors lt7	355-95-3	DTXSID70379270	210.10	C6H5F7	3	F	2.45 × 10^3^	1000
2-(Perfluorohexyl)ethanethiol	6:2 FtSH	Other aliphatics lt7	34451-26-8	DTXSID20379947	380.17	C8H5F13S	6	P	2.01 × 10^0^	1000
2-(Perfluorohexyl)ethyl methacrylate	6:2 FTMAc	Fluorotelomer PFAA precursors lt7	2144-53-8	DTXSID3047558	432.18	C12H9F13O2	6	P	4.32 × 10^−2^	1000
2-(Perfluorooctyl)ethanthiol		Other aliphatics gte7	34143-74-3	DTXSID20337446	480.18	C10H5F17S	8	P	4.90 × 10^−2^	1000
2-(Perfluorooctyl)ethyl acrylate	8:2 FTAc	Fluorotelomer PFAA precursors gte7	27905-45-9	DTXSID5067348	518.17	C13H7F17O2	8	P	1.53 × 10^−1^	1000
2-(Perfluorooctyl)ethyl methacrylate	8:2 FTMAc	Fluorotelomer PFAA precursors gte7	1996-88-9	DTXSID8062101	532.20	C14H9F17O2	8	P	2.20 × 10^−2^	1000
2-(Trifluoromethoxy)ethyl trifluoromethanesulfonate		other	329710-76-1	DTXSID00442840	262.12	C4H4F6O4S	1	F	4.18 × 10^−1^	1000
2,2-Difluoroethyl triflate		other	74427-22-8	DTXSID30378880	214.11	C3H3F5O3S	1	F	1.94 × 10^1^	1000
2-Amino-2H-perfluoropropane		other	1619-92-7	DTXSID70481246	167.05	C3H3F6N	1	F	1.36 × 10^2^	1000
2-Aminohexafluoropropan-2-ol		other	31253-34-6	DTXSID80382093	183.05	C3H3F6NO	1	F	2.45 × 10^−1^	1000
2H-Perfluoroisopropyl 2-fluoroacrylate		other	74359-06-1	DTXSID30622698	240.08	C6H3F7O2	1	F	2.88 × 10^2^	1000
3-(Perfluoro-2-butyl)propane-1,2-diol		Other aliphatics lt7	125070-38-4	DTXSID10382147	294.12	C7H7F9O2	4	F	4.22 × 10^−2^	1000
3-(Perfluoro-3-methylbutyl)-1,2-propenoxide		Other aliphatics lt7	54009-81-3	DTXSID00379884	326.11	C8H5F11O	2	P	4.98 × 10^1^	1000
3-(Perfluoroheptyl)propanoic acid	7:3 FTCA	Fluorotelomer PFAA precursors gte7	812-70-4	DTXSID90382620	442.12	C10H5F15O2	7	F	5.47 × 10^−3^	1000
3-(Perfluorohexyl)-1,2-epoxypropane		Other aliphatics lt7	38565-52-5	DTXSID30880413	376.12	C9H5F13O	6	F	2.95 × 10^1^	1000
3-(Perfluoroisopropyl)-2-propenoic acid		Fluorotelomer PFAA precursors lt7	243139-64-2	DTXSID40380257	240.08	C6H3F7O2	1	P	2.68 × 10^0^	1000
3-(Perfluorooctyl)propanol	8:3 FTOH	Other aliphatics gte7	1651-41-8	DTXSID10379991	478.15	C11H7F17O	8	P	7.39 × 10^−2^	1000
3-(Perfluoropropyl)propanol		Other aliphatics lt7	679-02-7	DTXSID60379269	228.11	C6H7F7O	3	F	1.70 × 10^1^	1000
3,3-Bis(trifluoromethyl)-2-propenoic acid		other	1763-28-6	DTXSID30170109	208.06	C5H2F6O2	1	P	2.55 × 10^0^	1000
3:1 Fluorotelomer alcohol		Fluorotelomer PFAA precursors lt7	375-01-9	DTXSID4059914	200.06	C4H3F7O	3	P	3.32 × 10^1^	1000
3H,3H-Perfluoro-2,4-hexanedione		Other aliphatics lt7	20825-07-4	DTXSID90174941	258.07	C6H2F8O2	2	F	1.25 × 10^3^	1000
3-Methoxyperfluoro(2-methylpentane)		Other aliphatics lt7	132182-92-4	DTXSID20881338	350.08	C7H3F13O	2	F	3.56 × 10^2^	1000
4,4-bis(Trifluoromethyl)-4-fluoropropanoic acid		Fluorotelomer PFAA precursors lt7	243139-62-0	DTXSID80380256	242.09	C6H5F7O2	1	P	1.74 × 10^0^	1000
4:2 Fluorotelomer alcohol	4:2 FTOH	Fluorotelomer PFAA precursors lt7	2043-47-2	DTXSID1062122	264.09	C6H5F9O	4	P	3.62 × 10^0^	1000
4:2 Fluorotelomer sulfonic acid	4:2 FTSA	Fluorotelomer PFAA precursors lt7	757124-72-4	DTXSID30891564	328.15	C6H5F9O3S	4	P	1.32 × 10^−6^	1000
4H-Perfluorobutanoic acid		Other aliphatics lt7	679-12-9	DTXSID50892417	196.05	C4H2F6O2	3	P	1.40 × 10^0^	1000
5H-Perfluoropentanal		Other aliphatics lt7	2648-47-7	DTXSID20337466	230.06	C5H2F8O	4	F	1.04 × 10^3^	1000
6:2 Fluorotelomer phosphate monoester	6:2 monoPAP	Fluorotelomer PFAA precursors lt7	57678-01-0	DTXSID90558000	444.09	C8H6F13O4P	6	P	1.32 × 10^−6^	1000
6:2 Fluorotelomer sulfonic acid	6:2 FTSA	Fluorotelomer PFAA precursors lt7	27619-97-2	DTXSID6067331	428.16	C8H5F13O3S	6	P	8.24 × 10^−7^	1000
6H-Perfluorohex-1-ene		Other aliphatics lt7	1767-94-8	DTXSID10379850	282.06	C6HF11	4	F	3.57 × 10^2^	1000
8:2 Fluorotelomer sulfonic acid	8:2 FTS	Fluorotelomer PFAA precursors gte7	39108-34-4	DTXSID00192353	528.18	C10H5F17O3S	8	P	1.00 × 10^−5^	1000
8H-Perfluorooctanoic acid	H-PFOA	Other aliphatics gte7	13973-14-3	DTXSID70565479	396.08	C8H2F14O2	7	P	5.35 × 10^−2^	1000
9H-Perfluorononanoic acid	H-PFNA	Other aliphatics gte7	76-21-1	DTXSID50226894	446.09	C9H2F16O2	8	P	2.62 × 10^−2^	1000
Allyl perfluoroisopropyl ether		Other aliphatics lt7	15242-17-8	DTXSID10370988	226.09	C6H5F7O	1	F	2.73 × 10^2^	1000
Ammonium perfluorooctanoate	PFOAA	PFAAs gte7	3825-26-1	DTXSID8037708	431.10	C8H4F15NO2	7	P	1.11 × 10^−1^	1000
Bis(1H,1H-perfluoropropyl)amine		Other aliphatics lt7	883498-76-8	DTXSID50381992	281.10	C6H5F10N	2	P	1.42 × 10^0^	1000
Dichloromethyl((perfluorohexyl)ethyl)silane		Silicon PFASs lt7	73609-36-6	DTXSID00223797	461.12	C9H7Cl2F13Si	6	F	1.13 × 10^−1^	1000
Dimethoxymethyl((perfluorohexyl)ethyl)silane		Silicon PFASs lt7	85857-17-6	DTXSID40235137	452.29	C11H13F13O2Si	6	P	8.37 × 10^−2^	1000
Ethyl pentafluoropropionyl acetate		Other aliphatics lt7	663-35-4	DTXSID20880144	234.12	C7H7F5O3	2	P	1.08 × 10^0^	1000
Ethyl perfluorobutyl ether		Other aliphatics lt7	163702-05-4	DTXSID0073118	264.09	C6H5F9O	4	F	1.89 × 10^2^	1000
Flurothyl		other	333-36-8	DTXSID5046516	182.07	C4H4F6O	1	F	1.57 × 10^2^	1000
Heptafluorobutyl iodide		Other aliphatics lt7	374-98-1	DTXSID4059912	309.95	C4H2F7I	3	P	1.07 × 10^2^	1000
Heptafluorobutyramide		Other aliphatics lt7	662-50-0	DTXSID2060965	213.06	C4H2F7NO	3	P	2.15 × 10^−1^	1000
Hexafluoroamylene glycol		Other aliphatics lt7	376-90-9	DTXSID3059927	212.09	C5H6F6O2	3	P	1.25 × 10^−1^	1000
Hexafluoroglutaryl chloride		Other aliphatics lt7	678-77-3	DTXSID0060985	276.94	C5Cl2F6O2	3	P	3.05 × 10^2^	1000
Methyl 3H-perfluoroisopropyl ether		Other aliphatics lt7	568550-25-4	DTXSID70537191	182.07	C4H4F6O	1	F	4.08 × 10^2^	1000
Methyl perfluoro(3-(1-ethenyloxypropan-2-yloxy)propanoate)	EVE	Other aliphatics lt7	63863-43-4	DTXSID8044969	422.10	C9H3F13O4	2	P	1.12 × 10^−1^	1000
Methyl perfluorohexanoate		Other aliphatics lt7	424-18-0	DTXSID20335700	328.08	C7H3F11O2	5	P	6.20 × 10^1^	1000
*N-Ethyl-N-*(2-hydroxyethyl)perfluorooctane sulfonamide	N-EtFOSE	FASA based PFAA precursors gte7	1691-99-2	DTXSID6027426	571.25	C12H10F17NO3S	8	P	8.78 × 10^−4^	1000
*N-Methyl-N-trimethylsilylheptafluorobutyramide*		Silicon PFASs lt7	53296-64-3	DTXSID40379666	299.26	C8H12F7NOSi	3	P	3.76 × 10^−1^	1000
Nonafluoropentanamide		Other aliphatics lt7	13485-61-5	DTXSID60400587	263.06	C5H2F9NO	4	P	4.96 × 10^0^	1000
Octafluoroadipamide		Other aliphatics lt7	355-66-8	DTXSID80310730	288.10	C6H4F8N2O2	4	P	5.99 × 10^−8^	1000
Pentadecafluorooctanoyl chloride		Other aliphatics gte7	335-64-8	DTXSID40187142	432.51	C8ClF15O	7	P	1.00 × 10^2^	1000
Pentafluoropropanoic anhydride		Other aliphatics lt7	356-42-3	DTXSID70870515	310.05	C6F10O3	2	F	1.24 × 10^2^	1000
Pentafluoropropionamide		Other aliphatics lt7	354-76-7	DTXSID0059871	163.05	C3H2F5NO	2	P	3.89 × 10^−1^	1000
Perfluamine	FTPA	Other aliphatics lt7	338-83-0	DTXSID9059834	521.07	C9F21N	3	F	1.94 × 10^2^	1000
Perfluoro-(2,5,8-trimethyl-3,6,9-trioxadodecanoic) acid		PFAAs lt7	65294-16-8	DTXSID70276659	662.10	C12HF23O5	3	F	7.55 × 10^−5^	1000
Perfluoro(4-methoxybutanoic acid)	PFMOBA	PFAAs lt7	863090-89-5	DTXSID60500450	280.05	C5HF9O3	3	P	1.75 × 10^0^	1000
Perfluoro(*N-methylmorpholine*)		Other aliphatics lt7	382-28-5	DTXSID7059933	299.04	C5F11NO	2	F	7.27 × 10^1^	1000
Perfluoro-1,3-dimethylcyclohexane		Non-PFAA perfluoroalkyls lt7	335-27-3	DTXSID0036926	400.06	C8F16	3	F	3.60 × 10^2^	1000
Perfluoro-1-iodohexane		PFAA precursors lt7	355-43-1	DTXSID7047566	445.95	C6F13I	6	P	2.88 × 10^2^	1000
Perfluoro-2,5-dimethyl-3,6-dioxanonanoic acid		PFAAs lt7	13252-14-7	DTXSID00892442	496.08	C9HF17O4	3	F	1.38 × 10^−3^	1000
Perfluoro-2-ethoxyethanesulfonic acid	PES	PFAAs lt7	113507-82-7	DTXSID50379814	316.09	C4HF9O4S	2	P	1.25 × 10^−6^	1000
Perfluoro-2-methyl-3-oxahexanoic acid	GenX	PFAAs lt7	13252-13-6	DTXSID70880215	330.05	C6HF11O3	3	F	2.41 × 10^−1^	1000
Perfluoro-3-(1H-perfluoroethoxy)propane	Freon E1	Other aliphatics lt7	3330-15-2	DTXSID8052017	286.04	C5HF11O	3	F	3.61 × 10^2^	1000
Perfluoro-3,6,9-trioxadecanoic acid		PFAAs lt7	151772-59-7	DTXSID80380837	412.06	C7HF13O5	2	P	3.20 × 10^−4^	1000
Perfluoro-3,6-dioxaheptanoic acid	PFECA B	PFAAs lt7	151772-58-6	DTXSID30382063	296.05	C5HF9O4	2	P	6.90 × 10^−4^	1000
Perfluoro-3,6-dioxaoctane-1,8-dioic acid		Other aliphatics lt7	55621-21-1	DTXSID20375106	322.06	C6H2F8O6	2	P	6.78 × 10^−5^	1000
Perfluoro-3-methoxypropanoic acid	PFMOPrA	PFAAs lt7	377-73-1	DTXSID70191136	230.04	C4HF7O3	2	P	6.93 × 10^−2^	1000
Perfluoro-4-isopropoxybutanoic acid	PFECA G	PFAAs lt7	801212-59-9	DTXSID60663110	380.06	C7HF13O3	3	P	4.35 × 10^−2^	1000
Perfluorobutanedioic acid		Other aliphatics lt7	377-38-8	DTXSID8059928	190.05	C4H2F4O4	2	P	9.24 × 10^−5^	1000
Perfluorobutanesulfonic acid	PFBS	PFAAs lt7	375-73-5	DTXSID5030030	300.09	C4HF9O3S	4	P	1.14 × 10^−8^	1000
Perfluorobutanesulfonyl fluoride	PFBS-F	PFAA precursors lt7	375-72-4	DTXSID20861913	302.09	C4F10O2S	4	F	3.93 × 10^3^	1000
Perfluorobutanoic acid	PFBA	PFAAs lt7	375-22-4	DTXSID4059916	214.04	C4HF7O2	3	P	3.36 × 10^1^	1000
Perfluorobutyraldehyde		PFAA precursors lt7	375-02-0	DTXSID10190946	198.04	C4HF7O	3	F	2.61 × 10^3^	1000
Perfluorocyclohexanecarbonyl fluoride		PFAA precursors lt7	6588-63-2	DTXSID80379781	328.06	C7F12O	5	F	6.10 × 10^1^	1000
Perfluoroglutaryl difluoride		Other aliphatics lt7	678-78-4	DTXSID50218052	244.04	C5F8O2	3	F	8.64 × 10^3^	1000
Perfluoroheptanoic acid	PFHpA	PFAAs lt7	375-85-9	DTXSID1037303	364.06	C7HF13O2	6	P	6.68 × 10^−2^	1000
Perfluoroheptanoyl chloride		Other aliphatics lt7	52447-22-0	DTXSID80382154	382.51	C7ClF13O	6	P	6.71 × 10^1^	1000
Perfluorohexanedioic acid		Other aliphatics lt7	336-08-3	DTXSID4059833	290.07	C6H2F8O4	4	P	7.28 × 10^−5^	1000
Perfluorohexanoic acid	PFHxA	PFAAs lt7	307-24-4	DTXSID3031862	314.05	C6HF11O2	5	P	9.03 × 10^−1^	1000
Perfluoromethylcyclopentane	PMCP	Non-PFAA perfluoroalkyls lt7	1805-22-7	DTXSID7061982	300.05	C6F12	4	F	2.58 × 10^2^	1000
Perfluorooct-1-ene		Non-PFAA perfluoroalkyls lt7	559-14-8	DTXSID40204489	400.06	C8F16	6	F	2.98 × 10^2^	1000
Perfluorooctanamide		Other aliphatics gte7	423-54-1	DTXSID60195123	413.09	C8H2F15NO	7	P	1.14 × 10^−1^	1000
Perfluorooctane		Non-PFAA perfluoroalkyls gte7	307-34-6	DTXSID0059794	438.06	C8F18	8	F	9.34 × 10^3^	1000
Perfluorooctanesulfonyl fluoride	PFOS-F	PFAA precursors gte7	307-35-7	DTXSID5027140	502.12	C8F18O2S	8	F	2.53 × 10^1^	1000
Perfluorooctanoic acid	PFOA	PFAAs gte7	335-67-1	DTXSID8031865	414.07	C8HF15O2	7	P	1.11 × 10^−1^	1000
Perfluorooctanoyl fluoride	PFOA-F	PFAA precursors gte7	335-66-0	DTXSID0059829	416.06	C8F16O	7	P	3.26 × 10^2^	1000
Perfluoropentanamide		Other aliphatics lt7	355-81-7	DTXSID70366226	245.07	C5H3F8NO	4	P	5.22 × 10^−2^	1000
Perfluoropentanedioic acid		Other aliphatics lt7	376-73-8	DTXSID8059926	240.06	C5H2F6O4	3	P	8.32 × 10^−5^	1000
Perfluoropentanoic acid	PFPeA	PFAAs lt7	2706-90-3	DTXSID6062599	264.05	C5HF9O2	4	P	6.62 × 10^0^	1000
Perfluoropropanoic acid	PPF	PFAAs lt7	422-64-0	DTXSID8059970	164.03	C3HF5O2	2	P	1.03 × 10^1^	1000
Perfluoropropyl trifluorovinyl ether	PPVE	Other aliphatics lt7	1623-05-8	DTXSID0061826	266.04	C5F10O	3	F	2.13 × 10^2^	1000
Perfluorosuccinic anhydride		Other aliphatics lt7	699-30-9	DTXSID6061022	172.04	C4F4O3	1	F	9.71 × 10^1^	1000
Perfluorotetradecanoic acid	PFTeDA	PFAAs gte7	376-06-7	DTXSID3059921	714.12	C14HF27O2	13	P	1.02 × 10^−3^	1000
Perfluorotridecanoic acid	PFTriDA	PFAAs gte7	72629-94-8	DTXSID90868151	664.11	C13HF25O2	12	P	6.60 × 10^−4^	1000
Potassium perfluorobutanesulfonate	KPFBS	PFAAs lt7	29420-49-3	DTXSID3037707	338.18	C4F9KO3S	4	P	1.14 × 10^−8^	1000
Potassium perfluorooctanoate	PFOA-K	PFAAs gte7	2395-00-8	DTXSID00880026	452.16	C8F15KO2	7	P	1.11 × 10^−1^	1000
Sevoflurane		other	28523-86-6	DTXSID8046614	200.06	C4H3F7O	1	F	1.94 × 10^2^	1000
Sodium perfluorooctanoate	PFOA-Na	PFAAs gte7	335-95-5	DTXSID40880025	436.05	C8F15NaO2	7	P	1.11 × 10^−1^	1000
Triethoxy((perfluorohexyl)ethyl)silane		Silicon PFASs lt7	51851-37-7	DTXSID1074915	510.37	C14H19F13O3Si	6	F	1.23 × 10^−1^	1000
tris(Trifluoroethoxy)methane		other	58244-27-2	DTXSID30395037	310.12	C7H7F9O3	1	P	4.55 × 10^−1^	1000

**Table 2 toxics-12-00501-t002:** **Comparison with other published PFAS zebrafish screens:** Developmental toxicity endpoint values (i.e., mortality and morphology) were compared among screening studies, and the results are presented. Provided is the chemical name and a CAS number used commonly across all four laboratories. For “Britton et al. BMC”, a benchmark concentration (BMC) as defined by benchmark response (BMR) is reported herein. A lower BMC indicates higher potency. For “[32], Morph.BMD10”, a benchmark concentration (BMD10) defined as a 10% change relative to the background response is reported (data from Ref. [32] Supplemental Table S3, Column F, Morph.BMD10). For “[34], Percent Mortality”, a percentage of mortality calculated for chemicals tested only at the nominal concentration of 5 μM is reported. (Data from Ref. [34]; Supplemental Table S4 Mortality (%).) For “[33], survival or nominal % 6–72 hpf”, a percentage of survival or percent normal at 6 dpf is presented; for simplicity, columns for percent survival and percent normal were combined, and if the values varied, the lowest was selected for use (Data from Ref. [33] Figure 1). Colors represent potency with red and bolded text being compounds with a BMC less than 1 μM for Britton and Truong, and active compounds for Han and Dasgupta. Yellow represents those with a BMC greater than 1 and less than 10 μM for Britton and Truong. Green represents compounds with a BMC greater than 10 μM for Britton and Truong. Dark gray indicates inactive compounds. Light gray indicates compounds not tested in other screens.

Chemical Name	CAS Number	Britton et al., 2024—Chemical QC	Britton et al., 2024—BMC (μM)	[32]—Morph.BMD10 (μM)	[34]—Percent Mortality	[33]—Percent Survival or Normal at 6–72 hpf
Perfluorooctanesulfonamide	754-91-6	P	**0.26**	2.88	**100**	**0**
N-Methylperfluorooctanesulfonamide	31506-32-8	P	**0.44**	28.98	**34.4**	Not Tested
((Perfluorooctyl)ethyl)phosphonic acid	80220-63-9	P	**0.58**	Inactive	Not Tested	Not Tested
Perfluoro-3,6,9-trioxatridecanoic acid	330562-41-9	P	1.30	9.78	Inactive	Inactive
Perfluorohexanesulfonamide	41997-13-1	P	1.45	9.76	Not Tested	Not Tested
1H,1H,6H,6H-Perfluorohexane-1,6-diol diacrylate	2264-01-9	P	1.52	12.07	Not Tested	Not Tested
1,6-Diiodoperfluorohexane	375-80-4	P	1.85	Not Tested	Not Tested	Not Tested
Perfluoropinacol	918-21-8	P	1.94	Not Tested	Not Tested	Not Tested
N-Ethylperfluorooctanesulfonamide	4151-50-2	P	2.37	43.70	Inactive	Not Tested
Perfluoroundecanoic acid	2058-94-8	P	2.74	55.82	Not Tested	Not Tested
Potassium perfluorooctanesulfonate	2795-39-3	P	2.77	11.02	Inactive	Not Tested
(Perfluorobutyryl)-2-thenoylmethane	559-94-4	P	3.11	54.32	Not Tested	Not Tested
6:2 Fluorotelomer alcohol	647-42-7	P	3.37	Inactive	Inactive	Not Tested
1H,1H,10H,10H-Perfluorodecane-1,10-diol	754-96-1	P	3.47	11.16	Not Tested	Not Tested
1-(Perfluorofluorooctyl)propane-2,3-diol	94159-84-9	F	4.42	18.86	Not Tested	Not Tested
9-Chloro-perfluorononanoic acid	865-79-2	P	6.38	20.97	Not Tested	Not Tested
N-Methyl-N-(2-hydroxyethyl)perfluorooctanesulfonamide	24448-09-7	F	7.06	69.09	Not Tested	Not Tested
Perfluorooctanesulfonic acid	1763-23-1	P	7.48	15.50	Inactive	Inactive
1H,1H,5H,5H-Perfluoro-1,5-pentanediol diacrylate	678-95-5	P	7.62	2.47	Not Tested	Not Tested
2,2,3,3-Tetrafluoropropyl acrylate	7383-71-3	F	7.81	31.89	Not Tested	Not Tested
8:2 Fluorotelomer acrylate	27854-30-4	P	8.20	Not Tested	Not Tested	Inactive
2H,2H,3H,3H-Perfluorooctanoic acid	914637-49-3	P	8.77	Inactive	Inactive	Inactive
1H,1H,7H-Perfluoroheptyl 4-methylbenzenesulfonate	424-16-8	P	8.97	Inactive	Not Tested	Not Tested
3H-Perfluoro-2,2,4,4-tetrahydroxypentane	77953-71-0	F	9.29	30.23	Inactive	Not Tested
Perfluorodecanoic acid	335-76-2	P	9.34	**0.22**	Not Tested	Not Tested
1H,1H,8H,8H-Perfluorooctane-1,8-diol	90177-96-1	P	10.37	Inactive	Not Tested	Not Tested
Perfluoro-1-octanesulfonyl chloride	423-60-9	P	10.42	Inactive	Not Tested	Not Tested
((2,2,3,3-Tetrafluoropropoxy)methyl)oxirane	19932-26-4	P	10.44	Inactive	Not Tested	Not Tested
1-Iodopentadecafluoroheptane	335-58-0	P	10.89	Inactive	Not Tested	Not Tested
11-H-Perfluoroundecanoic acid	1765-48-6	P	11.17	22.11	Not Tested	Not Tested
8:2 Fluorotelomer methacrylate	52447-23-1	P	13.10	Not Tested	Not Tested	Not Tested
3-(Perfluorooctyl)propanol	375-50-8	P	14.80	Not Tested	Not Tested	Not Tested
Perfluoroheptanesulfonic acid	375-92-8	P	15.55	35.16	Not Tested	Not Tested
Perfluorooctanamidine	307-31-3	P	19.45	Inactive	Not Tested	Not Tested
Perfluorooctanesulfonamido ammonium iodide	1652-63-7	P	19.75	74.00	Inactive	Not Tested
2-(Perfluorobutyl)ethyl acrylate	52591-27-2	P	20.74	Inactive	Not Tested	Not Tested
1H,1H-Perfluoro-3,6,9-trioxadecan-1-ol	147492-57-7	F	21.25	Inactive	Not Tested	Not Tested
2-(Perfluorohexyl)ethylphosphonic acid	252237-40-4	P	21.90	Inactive	Not Tested	Not Tested
(Heptafluorobutanoyl)pivaloylmethane	17587-22-3	P	22.97	Inactive	Not Tested	Not Tested
Perfluorononanoic acid	375-95-1	P	25.45	Inactive	Inactive	Inactive
4:4 Fluorotelomer alcohol	3792-02-7	P	25.53	Inactive	Inactive	Not Tested
Dimethoxymethyl((perfluorohexyl)ethyl)silane	355-47-5	P	27.23	Not Tested	Not Tested	Not Tested
Perfluoro(2-(2-propoxypropoxy)-1H,1H-propan-1-ol)	14548-74-4	F	27.37	Not Tested	Not Tested	Not Tested
1-Iodo-1H,1H,2H,2H-perfluorononane	355-93-1	P	27.89	Not Tested	Not Tested	Not Tested
1-Iodo-1H,1H,2H,2H-perfluoroheptane	1682-31-1	P	29.34	Inactive	Not Tested	Not Tested
2,2,2-Trifluoroethyl perfluorobutanesulfonate	79963-95-4	P	29.36	Inactive	Not Tested	Not Tested
Perfluorohexanesulfonic acid	355-46-4	P	30.15	65.28	Not Tested	Not Tested
5H-Perfluoropentanal	330562-44-2	P	31.27	10.53	Not Tested	Not Tested
Potassium perfluorohexanesulfonate	3871-99-6	P	31.30	76.85	Inactive	Inactive
7:3 Fluorotelomer alcohol	25600-66-2	P	38.32	Inactive	Not Tested	Not Tested
6:1 Fluorotelomer alcohol	375-82-6	P	73.35	Inactive	Not Tested	Not Tested
Perfluamine	137780-69-9	P	75.28	Not Tested	Not Tested	Inactive
1H,1H-Heptafluorobutyl epoxide	423-49-4	P	79.17	Not Tested	Not Tested	Not Tested
3:3 Fluorotelomer carboxylic acid	356-02-5	P	87.79	Inactive	Not Tested	Inactive
1H,1H-Heptafluorobutanol	678-39-7	P	89.03	Inactive	Inactive	Not Tested
Perfluorobutyraldehyde	423-54-1	P	Inactive	41.48	Not Tested	Not Tested
(Perfluoro-5-methylhexyl)ethyl 2-methylprop-2-enoate	72629-94-8	P	Inactive	41.56	Not Tested	Not Tested
Perfluoro-1,3-dimethylcyclohexane	678-77-3	P	Inactive	50.87	Not Tested	Not Tested
3-(Perfluoroisopropyl)-2-propenoic acid	39108-34-4	P	Inactive	55.68	Not Tested	Inactive
Perfluoro-4-isopropoxybutanoic acid	63863-43-4	P	Inactive	58.77	Not Tested	Not Tested
1-Propenylperfluoropropane	57678-01-0	P	Inactive	72.20	Not Tested	Not Tested
(Heptafluoropropyl)trimethylsilane	307-35-7	F	Inactive	86.05	Not Tested	Not Tested
Perfluorotridecanoic acid	151772-58-6	P	Inactive	Inactive	Inactive	Inactive
Perfluorobutanoic acid	863090-89-5	P	Inactive	Inactive	Inactive	Inactive
Octafluoroadipamide	335-67-1	P	Inactive	Inactive	Inactive	Inactive
Perfluorooctanoic acid	55621-21-1	P	Inactive	Inactive	Inactive	Inactive
Potassium perfluorobutanesulfonate	13252-13-6	F	Inactive	Inactive	Inactive	Inactive
Perfluoro-3,6-dioxaoctane-1,8-dioic acid	307-24-4	P	Inactive	Inactive	Inactive	Inactive
1H,1H,2H,2H-Perfluorohexyl iodide	375-22-4	P	Inactive	Inactive	Inactive	Inactive
8:2 Fluorotelomer alcohol	29420-49-3	P	Inactive	Inactive	Inactive	Inactive
Perfluoro(4-methoxybutanoic) acid	757124-72-4	P	Inactive	Inactive	Inactive	Inactive
Perfluorooct-1-ene	375-85-9	P	Inactive	Inactive	Not Tested	Inactive
3,3-Bis(trifluoromethyl)-2-propenoic acid	377-73-1	P	Inactive	Inactive	Not Tested	Inactive
Hexafluoroglutaryl chloride	13252-14-7	F	Inactive	Inactive	Not Tested	Inactive
tris(Trifluoroethoxy)methane	2706-90-3	P	Inactive	Inactive	Not Tested	Inactive
Pentafluoropropanoic anhydride	65294-16-8	F	Inactive	Inactive	Not Tested	Inactive
Sevoflurane	336-08-3	P	Inactive	Inactive	Not Tested	Inactive
Perfluorocyclohexanecarbonyl fluoride	422-64-0	P	Inactive	Inactive	Not Tested	Inactive
Perfluorooctane	377-38-8	P	Inactive	Inactive	Not Tested	Inactive
Perfluoro-1-iodohexane	355-81-7	P	Inactive	Inactive	**17.8**	Not Tested
Perfluoro(N-methylmorpholine)	31253-34-6	F	Inactive	Inactive	Inactive	Not Tested
3H,3H-Perfluoro-2,4-hexanedione	355-66-8	P	Inactive	Inactive	Inactive	Not Tested
Perfluorooctanamide	374-40-3	F	Inactive	Inactive	Inactive	Not Tested
Perfluoroglutaryl difluoride	375-73-5	P	Inactive	Inactive	Inactive	Not Tested
1H,1H,5H-Perfluoropentanol	243139-64-2	P	Inactive	Inactive	Inactive	Not Tested
3-(Perfluorohexyl)-1,2-epoxypropane	129301-42-4	P	Inactive	Inactive	Inactive	Not Tested
2,2,3,3,4,4,4-Heptafluorobutyl methacrylate	662-50-0	P	Inactive	Inactive	Inactive	Not Tested
3-(Perfluoro-2-butyl)propane-1,2-diol	1763-28-6	P	Inactive	Inactive	Inactive	Not Tested
Flurothyl	883498-76-8	P	Inactive	Inactive	Inactive	Not Tested
1,1,1,3,3-Pentafluorobutane	74427-22-8	F	Inactive	Inactive	Inactive	Not Tested
4:2 Fluorotelomer sulfonic acid	355-27-1	F	Inactive	Inactive	Inactive	Not Tested
Perfluoropropanoic acid	375-01-9	P	Inactive	Inactive	Inactive	Not Tested
8:2 Fluorotelomer sulfonic acid	355-80-6	P	Inactive	Inactive	Inactive	Not Tested
Tetrafluorosuccinic acid	1691-99-2	P	Inactive	Inactive	Inactive	Not Tested
3-Perfluoroheptylpropanoic acid	679-02-7	F	Inactive	Inactive	Inactive	Not Tested
1H,2H-Hexafluorocyclopentene	13485-61-5	P	Inactive	Inactive	Inactive	Not Tested
Perfluorotetradecanoic acid	423-65-4	P	Inactive	Inactive	Inactive	Not Tested
Perfluoro-3,6-dioxaheptanoic acid	125070-38-4	F	Inactive	Inactive	Inactive	Not Tested
Perfluoroheptanoic acid	58244-27-2	P	Inactive	Inactive	Inactive	Not Tested
Perfluoro-3-methoxypropanoic acid	329710-76-1	F	Inactive	Inactive	Inactive	Not Tested
Perfluoro-2-methyl-3-oxahexanoic acid	335-99-9	P	Inactive	Inactive	Inactive	Not Tested
Perfluoropentanoic acid	2144-53-8	P	Inactive	Inactive	Inactive	Not Tested
Octafluoroadipic acid	2043-47-2	P	Inactive	Inactive	Inactive	Not Tested
Perfluorohexanoic acid	376-90-9	P	Inactive	Inactive	Inactive	Not Tested
Perfluoro-(2,5,8-trimethyl-3,6,9-trioxadodecanoic)acid	679-12-9	P	Inactive	Inactive	Inactive	Not Tested
Perfluoro-2,5-dimethyl-3,6-dioxanonanoic acid	3825-26-1	P	Inactive	Inactive	Inactive	Not Tested
Allyl perfluoroisopropyl ether	812-70-4	F	Inactive	Inactive	Not Tested	Not Tested
N-Ethyl-N-(2-hydroxyethyl)perfluorooctanesulfonamide	376-06-7	P	Inactive	Inactive	Not Tested	Not Tested
3-(Perfluoropropyl)propanol	13695-31-3	P	Inactive	Inactive	Not Tested	Not Tested
Bis(1H,1H-perfluoropropyl)amine	21652-58-4	F	Inactive	Inactive	Not Tested	Not Tested
Perfluoropentanamide	38565-52-5	F	Inactive	Inactive	Not Tested	Not Tested
2-(Trifluoromethoxy)ethyl trifluoromethanesulfonate	374-98-1	P	Inactive	Inactive	Not Tested	Not Tested
1H,1H,7H-Dodecafluoro-1-heptanol [Dodecafluoroheptanol]	2043-55-2	P	Inactive	Inactive	Not Tested	Not Tested
6:2 Fluorotelomer methacrylate	4180-26-1	P	Inactive	Inactive	Not Tested	Not Tested
Perfluorooctanesulfonyl fluoride	20825-07-4	F	Inactive	Inactive	Not Tested	Not Tested
1H,1H-Perfluorooctyl acrylate	355-43-1	P	Inactive	Inactive	Not Tested	Not Tested
4:2 Fluorotelomer alcohol	559-14-8	F	Inactive	Inactive	Not Tested	Not Tested
2-(Perfluorooctyl)ethanthiol	27905-45-9	P	Inactive	Inactive	Not Tested	Not Tested
Hexafluoroamylene glycol	338-83-0	F	Inactive	Inactive	Not Tested	Not Tested
2-Amino-2H-perfluoropropane	6588-63-2	F	Inactive	Inactive	Not Tested	Not Tested
Nonafluoropentanamide	307-34-6	F	Inactive	Inactive	Not Tested	Not Tested
Triethoxy((perfluorohexyl)ethyl)silane	50836-66-3	P	Inactive	Inactive	Not Tested	Not Tested
1-Bromopentadecafluoroheptane	801212-59-9	P	Inactive	Inactive	Not Tested	Not Tested
3-Methoxyperfluoro(2-methylpentane)	3834-42-2	F	Inactive	Inactive	Not Tested	Not Tested
4H-Perfluorobutanoic acid	307-98-2	P	Inactive	Inactive	Not Tested	Not Tested
Ammonium perfluorooctanoate	34143-74-3	P	Inactive	Inactive	Not Tested	Not Tested
Perfluoro(propyl vinyl ether) [Heptafluoropropyltrifluorovinyl ether]	51851-37-7	F	Inactive	Inactive	Not Tested	Not Tested
11:1 Fluorotelomer alcohol	375-88-2	F	Inactive	Inactive	Not Tested	Not Tested
1H,1H,8H,8H-Perfluoro-3,6-dioxaoctane-1,8-diol	27619-97-2	P	Inactive	Inactive	Not Tested	Not Tested
Perfluorobutanesulfonic acid	422-63-9	F	Inactive	Inactive	Not Tested	Not Tested
1-Pentafluoroethylethanol	663-35-4	P	Inactive	Inactive	Not Tested	Not Tested
Perfluoromethylcyclopentane	3934-23-4	P	Inactive	Inactive	Not Tested	Not Tested
2,2-Difluoroethyl triflate	335-95-5	P	Inactive	Inactive	Not Tested	Not Tested
6:2 Fluorotelomer sulfonic acid	73609-36-6	F	Inactive	Inactive	Not Tested	Not Tested
Methyl perfluoro(3-(1-ethenyloxypropan-2-yloxy)propanoate)	1651-41-8	P	Inactive	Inactive	Not Tested	Not Tested
Perfluorononanoyl chloride	1765-92-0	P	Inactive	Inactive	Not Tested	Not Tested
Perfluoro-1,4-diiodobutane	354-76-7	P	Inactive	Inactive	Not Tested	Not Tested
Perfluoroheptanoyl chloride	133331-77-8	P	Inactive	Inactive	Not Tested	Not Tested
1,6-Dibromododecafluorohexane	2395-00-8	P	Inactive	Inactive	Not Tested	Not Tested
1H,1H-Perfluoroheptylamine	2043-52-9	P	Inactive	Inactive	Not Tested	Not Tested
Perfluorooctanoyl fluoride	85857-17-6	P	Inactive	Inactive	Not Tested	Not Tested
Pentadecafluorooctanoyl chloride	54009-81-3	P	Inactive	Inactive	Not Tested	Not Tested
8H-Perfluorooctanoic acid	376-73-8	P	Inactive	Inactive	Not Tested	Not Tested
1H,1H,5H-Perfluoropentyl methacrylate	568550-25-4	F	Inactive	Inactive	Not Tested	Not Tested
1H,1H-Perfluorononylamine	53296-64-3	P	Inactive	Inactive	Not Tested	Not Tested
Ethyl perfluorobutyl ether	151772-59-7	P	Inactive	Not Tested	Not Tested	Inactive
Perfluoro-3-(1H-perfluoroethoxy)propane	678-78-4	F	Inactive	Not Tested	Inactive	Not Tested
1H,1H,9H-Perfluorononyl acrylate	356-42-3	F	Inactive	Not Tested	Inactive	Not Tested
Perfluorosuccinic anhydride	19430-93-4	F	Inactive	Not Tested	Inactive	Not Tested
Perfluorobutanesulfonyl fluoride	424-18-0	P	Inactive	Not Tested	Inactive	Not Tested
3,3,4,4,5,5,6,6,6-Nonafluorohexene [1H,1H,2H-Perfluoro1-hexene]	406-58-6	F	Inactive	Not Tested	Inactive	Not Tested
Heptafluorobutyl iodide	2648-47-7	F	Inactive	Not Tested	Inactive	Not Tested
2-Aminohexafluoropropan-2-ol	355-95-3	F	Inactive	Not Tested	Inactive	Not Tested
Methyl perfluorohexanoate	15242-17-8	F	Inactive	Not Tested	Inactive	Not Tested
Heptafluorobutyramide	1619-92-7	F	Inactive	Not Tested	Inactive	Not Tested
1,1,1,5,5,5-Hexafluoroacetylacetone	375-72-4	F	Inactive	Not Tested	Inactive	Not Tested
1H,1H,2H-Perfluoro-1-decene	1623-05-8	F	Inactive	Not Tested	Inactive	Not Tested
1H,1H-Perfluoropentylamine	333-36-8	F	Inactive	Not Tested	Inactive	Not Tested
6H-Perfluorohex-1-ene	28523-86-6	F	Inactive	Not Tested	Inactive	Not Tested
Perfluoro-3,6,9-trioxadecanoic acid	1767-94-8	F	Inactive	Not Tested	Inactive	Not Tested
4,4,5,5,6,6,7,7,8,8,9,9,9-Tridecafluorononanoic acid	163702-05-4	F	Inactive	Not Tested	Inactive	Not Tested
Perfluoro-3,6-dioxadecanoic acid	375-02-0	F	Inactive	Not Tested	Inactive	Not Tested
1H,1H-Perfluorooctylamine	307-29-9	P	Inactive	Not Tested	Not Tested	Not Tested
9-H-Perfluorononanoic acid	76-21-1	P	Inactive	Not Tested	Not Tested	Not Tested
2-(Perfluorohexyl)ethanethiol	1005-73-8	F	Inactive	Not Tested	Not Tested	Not Tested
4,4-bis(Trifluoromethyl)-4-fluoropropanoic acid	1522-22-1	-	Inactive	Not Tested	Not Tested	Not Tested
1H-Perfluoro-1,1-propanediol	699-30-9	F	Inactive	Not Tested	Not Tested	Not Tested
Perfluoro(2-ethoxyethane)sulfonic acid	3330-15-2	F	Inactive	Not Tested	Not Tested	Not Tested
Ethyl pentafluoropropionyl acetate	382-28-5	F	Inactive	Not Tested	Not Tested	Not Tested
2,2,3,3,4,4,5,5,6,6,7,7,8,8,8-Pentadecafluorooctyl methacrylate	335-27-3	F	Inactive	Not Tested	Not Tested	Not Tested
Sodium perfluorooctanoate	132182-92-4	F	Inactive	Not Tested	Not Tested	Not Tested
Dichloromethyl((perfluorohexyl)ethyl)silane	1805-22-7	F	Inactive	Not Tested	Not Tested	Not Tested
6:2 Fluorotelomer phosphate monoester	52447-22-0	P	Inactive	Not Tested	Not Tested	Not Tested
1H,1H,11H,11H-Perfluorotetraethylene glycol	918-22-9	P	Inactive	Not Tested	Not Tested	Not Tested
Pentafluoropropionamide	335-66-0	P	Inactive	Not Tested	Not Tested	Not Tested
1-(Perfluorohexyl)octane	335-64-8	P	Inactive	Not Tested	Not Tested	Not Tested
Potassium perfluorooctanoate	13973-14-3	P	Inactive	Not Tested	Not Tested	Not Tested
3-(Perfluoro-3-methylbutyl)-1,2-propenoxide	34451-26-8	P	Inactive	Not Tested	Not Tested	Not Tested
Hexafluoroglutaric acid	243139-62-0	P	Inactive	Not Tested	Not Tested	Not Tested
Methyl 3H-perfluoroisopropyl ether	113507-82-7	P	Inactive	Not Tested	Not Tested	Not Tested
N-Methyl-N-trimethylsilylheptafluorobutyramide	1996-88-9	P	Inactive	Not Tested	Not Tested	Not Tested
2H-Perfluoroisopropyl 2-fluoroacrylate	74359-06-1	F	Inactive	Not Tested	Not Tested	Not Tested

## Data Availability

All raw data are included in the Appendix A and will also be uploaded to https://doi.org/10.23645/epacomptox.6062623 in Fall of 2024.

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
