# Peer review of "Using Zebrafish to Screen Developmental Toxicity of Per- and Polyfluoroalkyl Substances (PFAS)"

_toxics, 2024, doi:10.3390/toxics12070501_

Round 1
Reviewer 1 Report
Comments and Suggestions for Authors
Britton et al. reported their screening of 182 unique PFAS chemicals for potential developmental toxicities with a zebrafish model. Overall, this is a well-conducted study, providing potentially important toxicological information regarding to many PFAS chemicals, especially for those without much toxicological information. There are, however, a few issues that should be addressed prior to publication.
1. Exposure initiation time. The exposure to PFAS chemicals started at 6-8 hours post fertilization. Which means that the embryos were not exposed for the first 6-8 hours, which could potentially decrease the efficacy of this screening. Please provide rationale of this initiation time point.
2. Evaluation time. It is critical to choose a good time point during development to observe for potenitial effects. In this study, evaluation was performed six days post fertilization. Which stage were the embryos in by that time? This stage is equivaluant to which stage of human embryo/fetus? Please provide rationale of selecting this evaluation time point.
3. The exposure concentrations. Please provide detailed tested concentrations for each chemical tested, and compare these concentrations with real-world exposure levels in the discussion.
4. Data interpretation. The observed endpoints are mostly gross morphological changes. While these are very strong evidences for developmental toxicity, embryos without such large-scale morphological changes do not necessarily exhibit no changes in the internal organs, metabolism patterns, etc, which are also developmental toxicities. So, it is recommended to state in discussion that this test system focuses on gross morphological effects, negative results do not necessarily grant no developmental toxicities in other forms, or change title to screeen gross-developmental toxicity / teratogenicity.
Author Response
Comment 1: Exposure initiation time. The exposure to PFAS chemicals started at 6-8 hours post fertilization. Which means that the embryos were not exposed for the first 6-8 hours, which could potentially decrease the efficacy of this screening. Please provide rationale of this initiation time point.
Response 1: Thank you for your comment. There are a couple of reasons we start the exposures at 6 hours post fertilization, which is around the beginning of gastrulation. (1) Our primary interest is human toxicity and extrapolation to human developmental exposure. In humans implantation of the fertilized egg usually takes place during the blastula to gastrula conversion, so that is when we would like to initiate the exposure. (2) We have previously collected and made publicly available on the EPA CompTox Dashboard larval zebrafish developmental toxicity data for over 1000 chemicals collected using the same protocol as in this present paper. In order to compare and contrast these PFAS data with those other compounds, it is important not to introduce new experimental variables.
Comment 2: Evaluation time. It is critical to choose a good time point during development to observe for potential effects. In this study, evaluation was performed six days post fertilization. Which stage were the embryos in by that time? This stage is equivalent to which stage of human embryo/fetus? Please provide rationale of selecting this evaluation time point.
Response 2: Thank you for your comment. On day 6 our zebrafish larvae are at the 4.3 standard length stage described in Parichy et al (Normal Table of Post-Embryonic Zebrafish Development: Staging by Externally Visible Anatomy of the Living Fish) Dev. Dyn. 2009 https://doi.org/10.1002/dvdy.22113). The larvae are past the swim bladder inflation stage (3.4 to 3.7 mm standard length) and not quite to the Early Flexion (4.4 to 4.7 mm standard length) stage (beginning of feeding). We know that our larvae will begin feeding on Day 7 post fertilization.
It is very difficult to equate a given phase of human development with zebrafish development as different systems develop with different time lines in the two species, but we do know that that the developmental pathways are very similar in all vertebrates, which is why zebrafish are considered a good model for vertebrate development (Horzmann and Freeman, Making Waves: New Developments in Toxicology with Zebrafish, Toxicol Sci. 2018, https://doi.org/10.1093/toxsci/kfy044; Nishimura et al, Using Zebrafish in Systems Toxicology for Developmental Toxicity Testing, Congenital Anomalies, 2015, https://doi.org/10.1111/cga.12142).
We choose 6 days for our assessments because we wanted the larvae to complete as much of early development as possible before entering the stage where the larvae required feeding; feeding complicates the experimental design considerably. In addition, we were cognizant of previous reports of the possible interaction between PFAS exposure and swim bladder inflation, so we wanted to choose a time point for assessment that was past the swim bladder inflation stage (which occurs at 4 to 5 days post fertilization in our hands) so that we could assess that endpoint.
Comment 3: The exposure concentrations. Please provide detailed tested concentrations for each chemical tested, and compare these concentrations with real-world exposure levels in the discussion.
Response 3: Thank you for your comment. All the exposure concentrations tested are listed in Supplemental Table 1 (Sheet 2 “Observations” Col F).The highest concentration tested was 100 uM. This is orders of magnitude higher than most environmental concentrations. The following was added at line #187 in the Materials and Methods: “All final exposure concentrations for each chemical are listed in Supplemental Table 1, Col F of the second sheet.”
This study was not designed to study the effects of environmental PFAS concentrations on fish development. As dose is a concentration x time consideration, studying the effects of ambient levels of PFAS should require chronic exposure of the embryos/larvae/fry. Instead, this study was part of a much larger effort described in the Environmental Protection Agency's National PFAS Testing Strategy to relate molecular structure to toxicity of a chosen group of 150 PFAS. The results for the present zebrafish developmental toxicity study described herein will be compared to many other studies (see list below published papers; others are to be finalized in the next 6 months) that used the same chemical stocks in the same concentration range as our study to gain insight into PFAS structural category toxicity profiles. Because the zebrafish work was part of that much larger effort, we were basically “locked into” the concentration ranges tested.
- Patlewicz, A. M. Richard, A. J. Williams, R. S. Judson, R. S. Thomas, Towards reproducible structure-based chemical categories for PFAS to inform and evaluate toxicity and toxicokinetic testing, Computational Toxicology 24 (2022)
- E. Carstens, T. Freudenrich, K. Wallace, S. Choo, A. Carpenter, M. Smeltz, M. S. Clifton, W. M. Henderson, A. M. Richard, G. Patlewicz, B. A. Wetmore, K. Paul Friedman, T. Shafer, Evaluation of Per- and Polyfluoroalkyl Substances (PFAS) In Vitro Toxicity Testing for Developmental Neurotoxicity, Chemical Research in Toxicology 36 (3) (2023) 402–419
- A. Houck, G. Patlewicz, A. M. Richard, A. J. Williams, M. A. Shobair, M. Smeltz, M. S. Clifton, B. Wetmore, A. Medvedev, Makarov, Bioactivity profiling of per- and polyfluoroalkyl substances (PFAS) identifies potential toxicity pathways related to molecular structure, Toxicology 457 (2021)
- A. Houck, K. P. Friedman, M. Feshuk, G. Patlewicz, M. Smeltz, M. S. Clifton, B. A. Wetmore, S. Velichko, A. Berenyi, E. L. Berg, Evaluation of 147 perfluoroalkyl substances for immunotoxic and other (patho)physiological activities through phenotypic screening of human primary cells, ALTEX 40 (2) (2023) 248–270.
- Kreutz, M. S. Clifton, W. M. Henderson, M. G. Smeltz, M. Phillips, J. F. Wambaugh, B. A. Wetmore, Category-Based Toxicokinetic Evaluations of Data-Poor Per- and Polyfluoroalkyl Substances (PFAS) using Gas Chromatography Coupled with Mass Spectrometry, Toxics 11 (5) (2023) 463.
- Smeltz, J. F. Wambaugh, B. A. Wetmore, Plasma Protein Binding Evaluations of Per- and Polyfluoroalkyl Substances f Category-Based Toxicokinetic Assessment, Chemical Research in Toxicology (May 2023).
- G. Smeltz, M. S. Clifton, W. M. Henderson, L. McMillan, B. A. Wetmore, Targeted Per- and Polyfluoroalkyl substances (PFAS) assessments for high throughput screening: Analytical and testing considerations to inform a PFAS stock quality evaluation framework, Toxicology and Applied Pharmacology 459 (2023).
- E. Stoker, J. Wang, A. S. Murr, J. R. Bailey, A. R. Buckalew, High-Throughput Screening of ToxCast PFAS Chemical Library for Potential Inhibitors of the Human Sodium Iodide Symporter, Chemical Research in Toxicology 36 (3) (2023)
- J. Degitz, J. H. Olker, J. S. Denny, P. P. Degoey, P. C. Hartig, M. C. Cardon, S. A. Eytcheson, J. T. Haselman, S. A. Mayasich, M. W. Hornung, In vitro screening of per- and polyfluorinated substances (pfas) for interference with seven thyroid hormone system targets across nine assays, Toxicology in Vitro 95 (2024)
Comment 4: Data interpretation. The observed endpoints are mostly gross morphological changes. While these are very strong evidences for developmental toxicity, embryos without such large-scale morphological changes do not necessarily exhibit no changes in the internal organs, metabolism patterns, etc, which are also developmental toxicities. So, it is recommended to state in discussion that this test system focuses on gross morphological effects, negative results do not necessarily grant no developmental toxicities in other forms, or change title to screen gross-developmental toxicity / teratogenicity.
Response 4: Thank you for your comment; that is an excellent point! The following verbiage was added to the end of the paragraph in the Discussion discussing Table 2 at line #465.
It should be noted that in the present study as well as in the other studies listed in Table 2, the endpoints employed could best be classified as gross morphological changes. Normal-appearing animals do not necessarily signal the lack of developmental toxicity for the test chemical. Internal anatomy, physiology, and function may have been affected by the PFAS exposure without affecting gross morphology.
Reviewer 2 Report
Comments and Suggestions for Authors
Britton et al. used zebrafish at embryonic and larval stages to investigate developmental toxicity of 185 PFAS. As these “forever” chemicals are almost omnipresent, a detailed analysis of their toxicity profile is very important. Here, the authors used several endpoints (morphological abnormalities), which they scored manually at 6dpf upon treating from 6hpf onwards without removing embryos from the chorion. They found around 30% of PFAS to cause developmental toxicity.
They also compared their results with 3 other publications, which showed some commonalities, but also discrepancies. Overall, this study adds to our knowledge of potential PFAS-related toxicities and hints at certain substructures with specific risk (e.g. sulfonyl and sulfonamide residues were associated with zebrafish developmental toxicities).
Points of criticism
1) The authors mention the excellent reproducibility of their data in the discussion.
However, they also mention that they only produced replicates for three compounds (page 7 line 293). It is unclear why they claim high reproducibility if only 3/185 compounds were tested in duplicate?
2) A surprising finding is that the purity of 30% of tested compounds was not sufficient to pass quality control in order to derive any toxicity insights. Typically, commercial vendors ensure a certain degree of purity so it is unclear to me, why this is not the case here and the authors suggest to re-analyse the compounds.
3) The authors did not dechorionate the embryos. The chorion might influence uptake of compounds. Why did the authors decide to keep the fish in the chorion?
4) The comparison with other published studies, where the chorion was removed allows for speculation, which compounds might not penetrate the chorion. The authors should consider looking into this and adding this to the discussion.
5) Line 250-252 standard derivation or standard deviation?
Author Response
Comment 1: The authors mention the excellent reproducibility of their data in the discussion.
However, they also mention that they only produced replicates for three compounds (page 7 line 293). It is unclear why they claim high reproducibility if only 3/185 compounds were tested in duplicate?
Response 1: Thank you for your comment. That is true, so we have removed that sentence from the first sentence of the discussion, but I would like to explain why we felt so positively about the reproducibility: there was excellent concordance between the results from the single-concentration screening and the retesting of many of the same chemicals in the multiple-concentration screening portion of the study. Since we did not, however, present a formal assessment of that comparison in this manuscript, we have elected to remove that sentence from the Discussion.
Comment 2: A surprising finding is that the purity of 30% of tested compounds was not sufficient to pass quality control in order to derive any toxicity insights. Typically, commercial vendors ensure a certain degree of purity so it is unclear to me, why this is not the case here and the authors suggest to re-analyse the compounds.
Response 2: Thank you for your comment. Yes, that level of quality control fails was also a surprise for us, too. In fact, those samples were tested multiple times as explained in the referenced paper [Smeltz et al, 2023, Toxicol Appl Pharmacol; Targeted Per- and Polyfluoroalkyl Substances (PFAS) Assessments for High Throughput Screening: Analytical and Testing Considerations to Inform a PFAS Stock Quality Evaluation Framework (https://doi.org/10.1016/j.taap.2022.116355]. Even if the supplier attests to the purity of the chemical, a lot happens between the vendor assessment, the preparation of the solutions and the testing of the chemicals. As can be seen in Figure 3, many of the QC fail chemicals were volatile and so may have been lost during the DMSO solubilization or partitioned into the head space of the solution vial. Additionally, it is known that some PFAS break down in DMSO.
Comment 3: The authors did not dechorionate the embryos. The chorion might influence uptake of compounds. Why did the authors decide to keep the fish in the chorion?
Response 3: Thank you for your comment. We decided to use chorionated embryos because (1) we wanted to use hatching as an endpoint as it correlates with the highest number of other developmental endpoints (Ducharme et al. Meta-analysis of Toxicity and Teratogenicity of 133 Chemicals from Zebrafish Developmental Toxicity Studies, 2013 https://doi.org/10.1016/j.reprotox.2013.06.070 ); (2) dechorionation is a stressful process (either enzymatically or mechanically) leading to higher incidence of death and malformations in the control embryos; (3) we are unaware of any evidence that PFAS is impeded by the chorion. Interestingly, if we compare Truong et al (who dechorionated their embryos) with our results, we found 21 chemicals that tested positive in our assay that were inactive in their assay, whereas only 7 chemicals were active in their assay and negative in ours; this does not seem to indicate that the chorion in an impediment for PFAS entry; (4) because we have previously collected and made publicly available on the EPA CompTox Dashboard larval zebrafish developmental toxicity data for over 1000 chemicals collected using the same protocol as in this present paper, to compare and contrast these PFAS data with those other compounds, it is important not to introduce new experimental variables.
Comment 4: The comparison with other published studies, where the chorion was removed allows for speculation, which compounds might not penetrate the chorion. The authors should consider looking into this and adding this to the discussion.
Response 4: Thank you for this comment. As mentioned above in our answer to comment #3, there doesn’t seem to be any indication that the chorionated embryos experienced less toxicity than the dechorionated embryos. Moreover, we can find nothing in the literature that implies that the chorion may serve as a barrier preventing the entry of PFAS chemicals into the zebrafish embryo. There is one paper (Zoodsma et al, 2024 Perfluorooctane Sulfonate (PFOS) Negatively Impacts Prey Capture Capabilities in Larval Zebrafish; https://doi.org/10.1002/etc.5819) that states that the “PFOS uptake was unaffected by the presence of a chorion.”
Comment 5: Line 250-252 standard derivation or standard deviation?
Response 5: Thank you for your comment. The reviewer is correct; it should have been “deviation.” That section at line #249 has now been rewritten as “In addition to estimated activity concentrations inducing a specified level of responses (e.g. 10%, 50%, etc.), a benchmark concentration (BMC) was also derived in tcpl using a specified benchmark response level (BMR) of 1.349 times the standard deviation of baseline (10%) [42]. Given lack of baseline variability in dichotomous observations, the baseline median absolute deviation was set as 5%.”